# circRNA432 enhances the coelomocyte phagocytosis via regulating the miR-2008-ELMO1 axis in *Vibrio splendidus*-challenged *Apostichopus japonicus*

Xianmu Fu[1], Ming Guo 🄳 [1✉], Jiqing Liu[1] & Chenghua Li 🄳 [1,2✉]

Circular RNAs (circRNAs) are a kind of extensive and diverse covalently closed circular endogenous RNA, which exert crucial functions in immune regulation in mammals. However, the functions and mechanisms of circRNAs in invertebrates are largely unclarified. In our previous work, 261 differentially expressed circRNAs including circRNA432 (circ432) were identified from skin ulcer syndrome (SUS) diseased sea cucumber *Apostichopus japonicus* by RNA-seq. To better address the functional role of sea cucumber circRNAs, circ432 was first found to be significantly induced by *Vibrio splendidus* challenge and LPS exposure in this study. Knock-down circ432 could depress the *V. splendidus*-induced coelomocytes phagocytosis. Moreover, circ432 is validated to serve as the sponge of miR-2008, a differential expressed miRNA in SUS-diseased sea cucumbers, by Argonaute 2-RNA immunoprecipitation (AGO2-RIP) assay, luciferase reporter assay and RNA fluorescence in situ hybridization (FISH) in vitro. Engulfment and cell motility protein 1 (AjELMO1) is further demonstrated to be the target of miR-2008, and silencing AjELMO1 inhibits the *V. splendidus*-induced coelomocytes phagocytosis, and this phenomenon could be further suppressed by supplementing with miR-2008 mimics, suggesting that circ432 might regulate coelomocytes phagocytosis via miR-2008-AjELMO1 axis. We further confirm that the depressed coelomocytes' phagocytosis by circ432 silencing is consistent with the decreased abundance of AjELMO1, and could be recovered by miR-2008 inhibitors transfection. All our results provide the evidence that circ432 is involved in regulating pathogen-induced coelomocyte phagocytosis via sponge miR-2008 and promotes the abundance of AjELMO1. These findings will enrich the regulatory mechanism of phagocytosis in echinoderm and provide theoretical data for SUS disease prevention and control in sea cucumbers.

[1] State Key Laboratory for Managing Biotic and Chemical Threats to the Quality and Safety of Agro-products, Ningbo University, 315211 Ningbo, P. R. China. [2] Laboratory for Marine Fisheries Science and Food Production Processes, Qingdao National Laboratory for Marine Science and Technology, 266071 Qingdao, P. R. China. ✉email: guoming@nbu.edu.cn; lichenghua@nbu.edu.cn

Phagocytosis is a cellular process that internalizes foreign particles (including bacteria, viruses, and fungi), which can eliminate microorganisms quickly and effectively[1]. In vertebrates, phagocytosis usually involves specialized phagocytes (such as macrophages and neutrophils)[2]. They enter the cell directly through the way of cell membrane collapse after the recognition of conserved motifs on the pathogen by receptors on the cell membrane, sometimes accompanied by the extension of the cell membrane[3]. Research supports that all invertebrates contain phagocytotic deformation cells[4]. In most invertebrates, some non-specialized phagocytes can also uptake and remove pathogenic microorganisms through phagocytosis. As a representative of marine invertebrates, coelomocytes of sea cucumber *Apostichopus japonicus*, as immune cells, can also play a role in eliminating pathogens by combining with immune factors in coelom fluid[5]. According to previous reports, sea cucumber coelomocytes are morphologically divided into four types of cells (spherocytes, amoebocytes, clear cells, and lymphoid cells), but only amoebocytes and spherocytes in sea cucumber have phagocytosis[6], and then degrade or expel foreign bodies in vitro[7]. To date, two functional proteins have been confirmed to be involved in this phagocytosis process in sea cucumbers. Integrin modulates coelomocyte phagocytosis via the activation of septin2/7[8]; NOD-like receptor family CARD-containing 4 protein (NLRC4), as a new membrane PRR, mediates coelomocyte phagocytosis and further clearance of intracellular *Vibrio* through the NLRC4-β-actin-Arp2/3 complex-lysosome pathway[9]. However, functional proteins regulating phagocytosis in sea cucumber coelomocytes are still far from fully understood.

Engulfment and cell motility protein 1 (ELMO1) is a transmembrane protein that plays an important role in cell migration and phagocytosis in vertebrates[10]. The N-terminal region of ELMO1 is essential for membrane targeting, while the C-terminal region interacts with proteins (such as Dock180) to regulate downstream molecules[11,12]. Although ELMO1 has no intrinsic catalytic activity, it mediates multiple cellular functions by providing scaffolds for signal proteins or regulating the activity of other proteins through protein–protein interactions[13]. In phagocytosis (endocytosis) regulation, ELMO1 functionally cooperates with Dock180 to regulate Rac1 activity and promote phagocytosis and cell shape changes in mammalian cells[14], such as inhibiting the expression of ELMO1/Dock180 complex diminished Rac1 activation and *Salmonella* phagocytosis in mouse intestinal epithelial cells[15]. However, as most of our insights into ELMO1 phagocytosis stem from studies of ELMO1 and Dock180, the mechanisms regulating ELMO1 to undergo phagocytosis remain largely unknown.

Non-coding RNAs (ncRNAs) are considered important regulators in gene or protein expression via affecting chromosome remodeling, transcription, translation, and protein function during the process of gene expression[16]. Circular RNAs (circRNAs) are one of the ncRNAs that play important roles in the regulation of gene expression at the post-transcriptional level[17]. circRNAs have been reported to have many important roles, including as protein sponges[18], in translation[19], and as microRNA (miRNA) sponges[20]. To date, circRNAs functioning as the competing endogenous RNA (ceRNA) have been most widely reported in cell development and pathogen regulation. According to recent studies, circRNAs mainly function as miRNA sponges through miRNA response elements (MREs), specific sequence motifs complementary to particular miRNAs, and act as ceRNAs to alleviate miRNA-mediated downstream inhibitory target mRNA[21]. At present, the research on the regulation mechanism of mammalian circRNAs acting as miRNAs sponges is becoming more and more complete and clear[22,23], and the research on this mechanism in vertebrates (especially teleost fish) and

invertebrates is also constantly being explored. For instance, circDtx1 downregulates teleost IRF3-mediated antiviral immune responses by inhibiting miR-15a-5p-dependent TRIF[24]; circRasGEF1B enhances antiviral immunity by regulating the miR-21-3p/MITA pathway in vertebrates[25]; circRNA75 and circRNA72 inhibit sea cucumber coelomocytes apoptosis as the sponge of miR-200 by targeting Tollip[26]. However, the function and regulatory mechanisms of circRNAs as miRNA molecular sponges in phagocytosis are still lacking.

Sea cucumber *A. japonicus*, is an important economic species in China[27]. Unfortunately, with the continuous expansion of aquaculture, the outbreak of various viral and bacterial diseases has resulted in the high infectivity and lethality of sea cucumbers, especially skin ulceration syndrome (SUS) caused by *Vibrio splendidus* (*V. splendidus*). This problem has received widespread attention[28]. Thus, an in-depth analysis of the immune defense mechanism of sea cucumbers is considered to be the fundamental way and effective strategy to achieve disease prevention. In our previous work, 261 differentially expressed circRNAs were shown in healthy and SUS-diseased sea cucumbers, of which 71.6% were intergenic circRNAs[29]. This indicates that most sea cucumber circRNAs were much longer, and this is different from other species such as humans and mice. Among them, circRNA432 (circ432) was also an intergenic circRNA, and its length is 2358 bp, which is much shorter than most other identified circRNAs. This is conducive to the subsequent verification of the function of circ432. Moreover, in the data of the RNA-seq, circ432 was significantly upregulated by 8.023-fold between SUS-diseased and healthy sea cucumbers, suggesting that circ432 plays a vital role in mediating pathogen invasion. In this study, we first identified circ432 as a circular RNA by RT-PCR and Sanger sequencing and then analyzed by flow cytometry whether it mediates the phagocytosis of sea cucumber coelomocytes in response to *V. splendidus*. Then, we detected the binding of circ432 to miR-2008 and the binding of miR-2008 to *A. japonicus* ELMO1 (AjELMO1) through luciferase report assay, RIP assay, or RNA-fluorescence in situ hybridization (FISH). Subsequently, the abundance levels of circ432, miR-2008, and *AjELMO1* were investigated in *V. splendidus*-challenged or Lipopolysaccharides (LPS)-exposed sea cucumber coelomocytes. Finally, functional validation of circ432, miR-2008, and AjELMO1 was conducted in vitro and in vivo to elucidate their connection. Our present work provided ideas for the immune-regulation mechanism of sea cucumbers under a pathogen challenge.

## Results

**Characterization of circ432 in sea cucumber coelomocytes.** We had previously investigated the differentially expressed circRNA in sea cucumbers by using RNA-seq technology. A total of 3592 circRNAs were identified, including 261 differentially expressed circRNAs involved in regulating the occurrence of diseases, 117 were upregulated and 144 were downregulated[29]. Among them, circ432 derived from AJAPscaffold432: 393652│396009 of *A. japonicus* genome[30] was found to be significantly upregulated in SUS-diseased sea cucumbers, and it was 2358 bp in length (NCBI Accession no. OP242379) (Supplementary Fig 1, in Supplementary Information). To confirm whether the head-to-tail splicing was the result of trans-splicing or genome rearrangement, we amplified circ432 by divergent primers and validated the head-to-tail splicing in the RT-PCR product of circ432 by Sanger sequencing (Fig. 1a). Then, we amplified circ432 with divergent primers and convergent primers with cDNA or gDNA as templates, and found that circ432 could only be detected in cDNA by using only divergent primers, but not in gDNA (Fig. 1b). The RNase R resistance assay further showed that circ432 was more

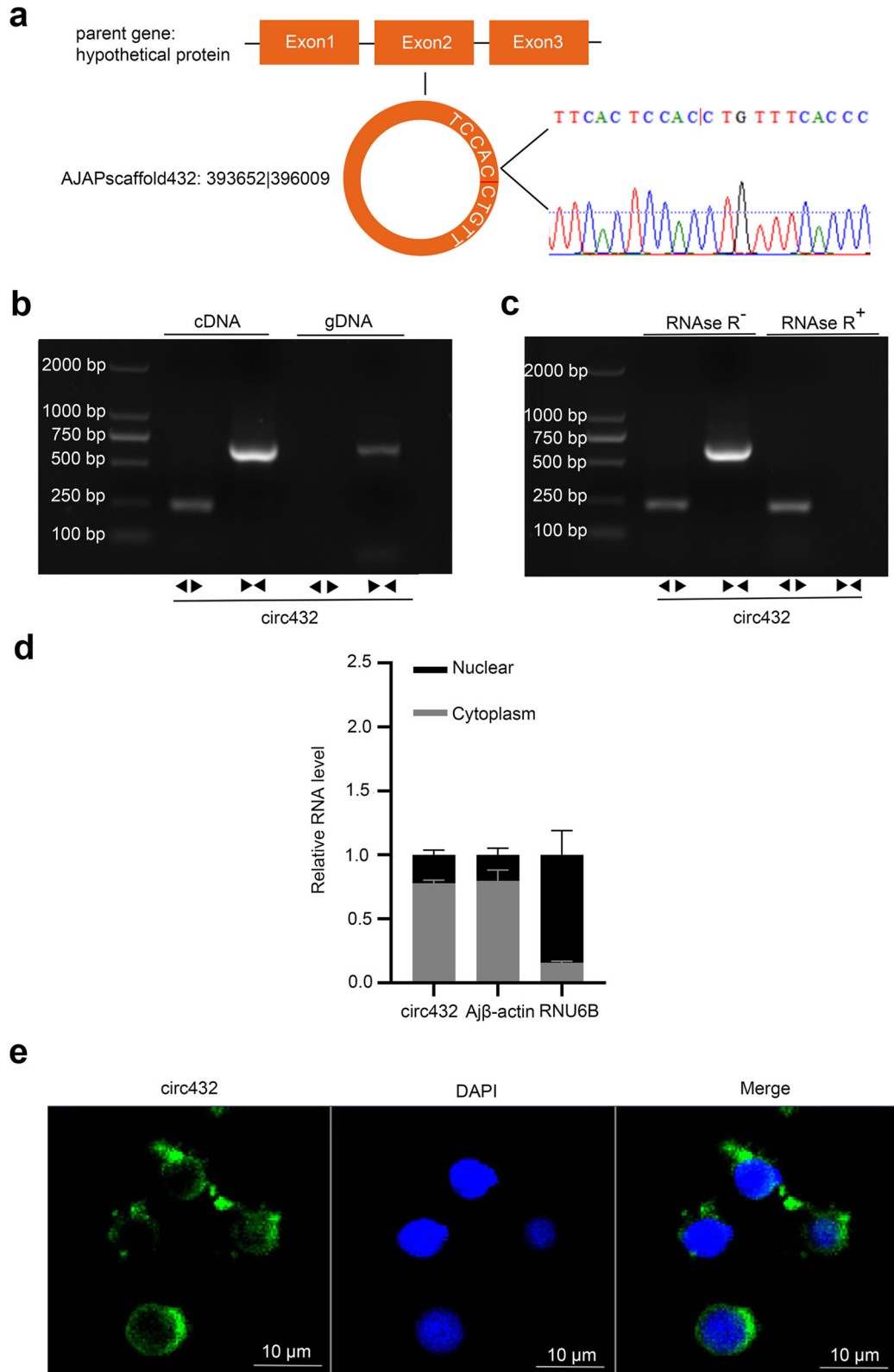

**Fig. 1 The characteristics of circ432. a** The junction splice site of circ432 was confirmed by Sanger sequencing. **b** RT-PCR with divergent (◄►) and convergent primers (►◄) showing the amplification of circ432 from cDNA or gDNA of coelomocytes. **c** The divergent and convergent primers of circ432 in the presence or absence of RNase R was validated by RT-PCR. **d** RNA extracted from the cytoplasm and nuclear of coelomocytes were performed to detect circ432 abundance by RT-PCR. Ajβ-actin, and RNU6B served as the controls, $n = 3$. **e** circ432 mainly located in the cytoplasm was verified by RNA FISH. Scale bar, 10 μm. All data represented the mean ± SD from three independent triplicated experiments. *$P < 0.05$, **$P < 0.01$.

resistant to RNase R digestion than the linear gene (Fig. 1c). RT-qPCR assay was validated in strict compliance with the Minimum Information for Publication of Quantitative Real-Time PCR Experiments (MIQE) guidelines (Supplementary Note 1, in Supplementary Information). To identify the most stable reference gene from healthy sea cucumber tissues and sea cucumber coelomocytes post-*V. splendidus* challenge for the normalization of qRT-PCR data, we detected the expression stability of ten housekeeping genes using the algorithm of geNorm, NormFinder, BestKeeper, and RefFinder, respectively. The results revealed that Ajβ-actin can be selected as a better-suited housekeeping gene in healthy sea cucumber tissues and sea cucumber coelomocytes post-*V. splendidus* infection, and the qRT-PCR data with Ajβ-actin as the reference gene is convincing in this study (Supplementary Note 2, in Supplementary Information). To address the spatial distribution of circ432, we detected circ432 in untreated coelomocytes by the cytoplasmic nuclear fractionation experiments and found that circ432 was mainly located in the cytoplasm (Fig. 1d). We also designed a specific fluorescent probe for circ432, and the RNA FISH result showed that circ432 was also mainly located in the cytoplasm of LPS-exposed coelomocytes (Fig. 1e). Taken together, these results indicated that circ432 was a stable expression of circRNA and mainly distributed in the cytoplasm.

**circ432 mediates coelomocytes phagocytosis in sea cucumber.** To verify the tissue-specific abundance of circ432, the abundance of circ432 in coelomocytes, muscle, respiratory trees, intestine, and tentacles was detected by qPCR (Fig. 2a). The result showed that the abundance level of circ432 in coelomocytes was the highest, followed by intestine and muscle, and the lowest abundance in the tentacles and respiratory tree. Additionally, to elucidate the immune function of circ432, qPCR was used to detect the abundance of circ432 in *V. splendidus*-challenged sea cucumbers and LPS-stimulated primary coelomocytes at various times points (Fig. 2b). In *V. splendidus*-challenged sea cucumbers, coelomocytes circ432 abundance was significantly induced from 24 to 72 h, and the peak value was detected at 48 h post-*V. splendidus* challenge. Similar to the abundance profile of *V. splendidus* challenge, the abundance of circ432 in LPS-stimulated coelomocytes was also significantly upregulated from 3 to 24 h, and the highest abundance occurred at 6 h post-stimulation. In sea cucumbers, coelomocytes are the main cell types involved in cellular immune function, such as proliferation, apoptosis, and phagocytosis[31]. To this end, we designed two siRNAs (si-circ432-1 and si-circ432-2) that specifically targeted circ432 to detect the immune function in response to *V. splendidus* infection (Table 1). Through in vivo and in vitro interference experiments, the results of the qPCR analysis showed that two siRNAs (si-circ432-1 and si-circ432-2) significantly decreased the abundance level of circ432, but this siRNAs do not affect the abundance level of the host gene of circ432 (Fig. 2c). Moreover, the abundance levels of circ432 were also detected post treated with different concentrations of si-circ432-1 and si-circ432-2. The results revealed that there was a dose-dependent relationship between the abundance level of circ432 and si-circ432-1 or si-circ432-2. However, it was found that the inhibition efficiency induced by si-circ432-1 was higher than that induced by si-circ432-2 (Supplementary Fig 2, in Supplementary Information), thus we selected si-circ432-1 for the subsequent experiments. Then, we transfected si-circ432-1 into each sea cucumber for 24 h and challenged with *V. splendidus* for another 24 h, and then assayed the coelomocytes' phagocytosis by flow cytometry. The results showed that the phagocytosis ratio in coelomocytes of si-circ432 + *V. splendidus*-treated sea cucumbers was significantly lower than that in si-NC + *V. splendidus*-treated

sea cucumbers (Fig. 2d). Collectively, our results confirmed that circ432 functions as a positive regulator involved in *V. splendidus*-induced sea cucumber coelomocyte phagocytosis.

**circ432 functions as a miRNA sponge of miR-2008.** Since circRNAs were predominantly located in the cytoplasm and usually function as miRNA sponges[24], we predicted whether circ432 could be bound with these 7 miRNAs by the miRanda program[32–38]. The results showed that circ432 only had potential binding sites with miR-2008, miR-9, and miR-137 (Supplementary Table 1, in Supplementary Information), but after the screening threshold was changed (fraction less than S > 120 (single residue fraction) and minimum free energy less than −17 kcal/mol), only 2 potential binding sites were predicted for circ432 and miR-2008 (Fig. 3a). To this end, we first performed RNA immunoprecipitation (RIP) for Argonaute 2 (AGO2) in coelomocytes to enrich circRNAs, and the results of RT-PCR indicated that endogenous circ432 could be pulled down by the AGO2 monoclonal antibody (Fig. 3b). Then, we constructed wild-type (circ432-wt) and mutant (circ432-mut-1, circ432-mut-2) circ432 luciferase plasmids containing the miR-2008 binding sites in the psiCHECK-2 vector (Table 1). After co-transfected Hela cells with luciferase plasmids and miR-2008M/NCM for 48 h, we found that miR-2008M could still inhibit the luciferase activity of the luciferase plasmids of circ432-mut-1 or circ432-mut-2 when only one site was mutated, but after mutated the sites of both circ432-mut-1 and circ432-mut-2 plasmids, miR-2008M could not affect their luciferase activity (Fig. 3c). Furthermore, we detected that miR-2008 mimics inhibited the luciferase activity of circ432 in a dose-dependent manner (Fig. 3c). At the cellular level, we synthesized a FITC-labeled probe to target circ432 and a Cy3-labeled probe to target miR-2008, and the results showed that circ432 and miR-2008 co-localized in the cytoplasm by RNA FISH in LPS-exposed coelomocytes (Fig. 3d). These results showed that circ432 might function as a sponge of miR-2008.

**miR-2008 inhibits phagocytosis of coelomocytes.** miR-2008 from sea cucumber had previously been confirmed to modulate reactive oxygen species production by targeting betaine-homocysteine S-methyltransferase[39], while the phagocytotic activities of miR-2008 have not been studied. To address this issue, miR-2008M or miR-2008I were transfected into coelomocytes to validate the immune function of miR-2008 (Table 1). Through in vivo and in vitro interference experiments, the qPCR results showed that miR-2008M significantly increased the abundance of miR-2008, while miR-2008I decreased the abundance of miR-2008 significantly (Fig. 4a). Then, miR-2008M and miR-2008I treatments were performed to detect the changes in the phagocytic capacity of sea cucumbers post-*V. splendidus* challenge. Through flow cytometry analysis, it was found that the phagocytosis ratio of coelomocytes treated with miR-2008M + *V. splendidus* was significantly lower than that of sea cucumbers treated with NCM + *V. splendidus* (Fig. 4b). On the contrary, the phagocytosis ratio in coelomocytes of miR-2008I + *V. splendidus*-treated sea cucumbers was significantly higher than that in NCI + *V. splendidus*-treated sea cucumbers (Fig. 4c). In general, our results demonstrated that miR-2008 mediated the phagocytosis of sea cucumber coelomocytes.

**miR-2008 targets AjELMO1 to inhibit coelomocytes phagocytosis in sea cucumber.** Given that miR-2008 mediates phagocytosis, we first screened out genes that might be related to phagocytosis in the transcriptome of *A. japonicus*[40], filtered some genes without complete sequence annotation, and then predicted the binding sites of miR-2008 and genes that might be related to

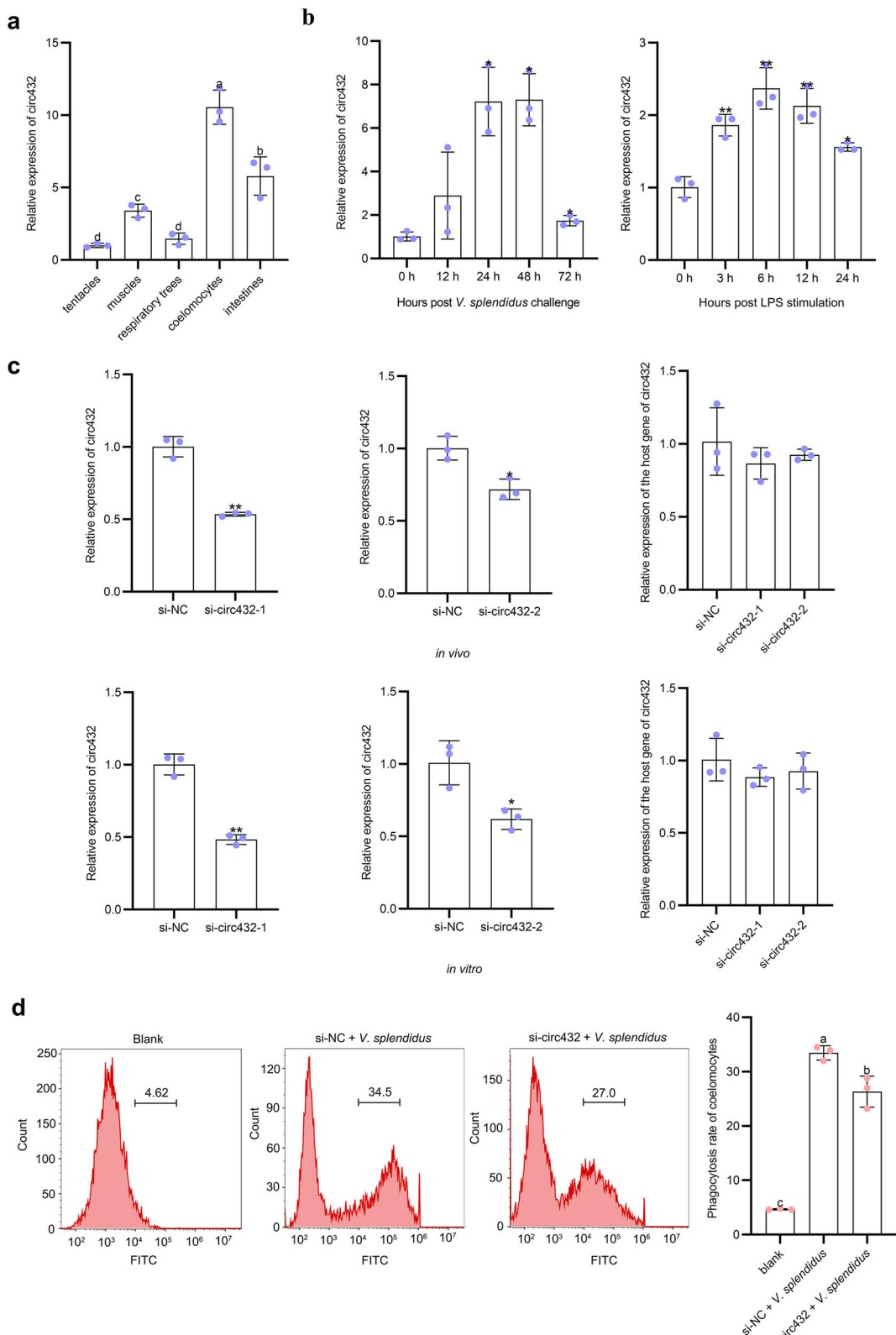

**Fig. 2 circ432 participates in the phagocytosis of coelomocytes induced by _V. splendidus_. a** Tissue abundance distribution of circ432 in coelomocytes, intestine, tentacles, respiratory tree, and muscle was measured by qPCR. **b** The time-course abundance patterns of circ432 in coelomocytes of _V. splendidus_-challenged _A. japonicus_ and the LPS-exposed primary cultured cell as measured by qPCR. **c** The relative abundance level of circ432 and the host gene of circ432 after circ432 siRNA transfection in vivo and in vitro were detected by qPCR. **d** The phagocytosis ratio of coelomocytes in circ432 siRNA transfection under the induction of _V. splendidus_ was detected by flow cytometry. All data represented the mean ± SD from three independent triplicated experiments. *$P < 0.05$, **$P < 0.01$. Different letters above each bar indicate significant differences: $P < 0.05$, whereas bars with the same letter indicates non-significant differences.

**Table 1 Primers used in this study.**

| Primer name | Primer sequence (5'-3') | Used for |
|---|---|---|
| circ432-div | CAGACTTCCCTGTGGTGA TGGTGTATGGACTGGCTC | Real-time PCR |
| circ432-con | TTGGATGTTGGATGCTCA CATTGGGTGGAACAGGTG | Real-time PCR |
| circ432 host gene | ACTTCCCAGTGGTAAATG ATCTCATTGGGTGGAACA | Real-time PCR |
| *AjELMO1*-3'-1 *AjELMO1*-3'-2 | TTATAGTCTGGTGGCGGTGGTGGG GTCAAACACTTCTTGCGAACTGGC | 3' RACE |
| *AjELMO1*-5'-1 *AjELMO1*-5'-2 | TCATCCACCCGAACACCTAACCCA CACCAGCCGTTCAGAGAAGACCGA | 5' RACE |
| *AjELMO1*-F-*Hind*III *AjELMO1*-R-*Xho*I | CCAAGCTTATGAGTCGGGCAGTA CCCTCGAGCTACGACTCGTAGTAGAAGT | Recombinant expression |
| Ajβ-actin | CCATTCAACCCTAAAGCCAACA ACACACCGTCTCCTGAGTCCAT | Real-time PCR |
| RNU6B | CGTGAAGCGTTCCATATTTTAA | Real-time PCR |
| UP2 | GAATCGAGCACCAGTTACGCAT | Real-time PCR |
| miR-2008 | AUCAGCCUCGCUGUCAAUACG | Real-time PCR |
| miR-2008 mimics | AUCAGCCUCGCUGUCAAUACG UAUUGACAGCGAGGCUGAUUU | miRNA overexpression |
| miR-2008 inhibitor | CGUAUUGACAGCGAGGCUGAU | miRNA inhibition |
| NCI | CAGUACUUUUGUGUAGUACAA | NC for miR-2008 inhibition |
| NCM | UUCUCCGAACGUGUCACGUTT ACGUGACACGUUCGGAGAATT | NC for miR-2008 overexpression |
| circ432-siRNA-1 | GGUCUUCACUCCACCUGUUTT AACAGGUGGAGUGAAGACCTT | RNA interference |
| circ432-siRNA-2 | CCACCUGUUUCACCCGAUGTT CAUCGGGUGAAACAGGUGGTT | RNA interference |
| *AjELMO1*-siRNA | GACGGACAAAGCGUUUGAATT UUCAAACGCUUUGUCCGUCTT | RNA interference |
| si-NC | UUCUCCGAACGUGUCACGUTT ACGUGACACGUUCGGAGAATT | NC for si-circ432 and si-ELMO1 |
| circ432-wt | CCCTCGAGAATCTCTTGCTTTTATAT GGGTTTAAACGAATTGGATTACTCACCC | Luciferase vector contruction |
| circ432-mut-1 | TATGCATAGTAGGGATGATTTTGAA TCCCTACTATGTCATACCAAAATATA | Luciferase vector contruction |
| circ432-mut-2 | TTTGACTTAGTAGGGATGATTTTGAA TCCCTACTAAGTCAAACAAAAATATT | Luciferase vector contruction |
| *AjELMO1*-wt | CCCTCGAGGACCAAGACCCTCTGTGA GGGTTTAAACAGAAGTATGTTGATGCCG | Luciferase vector contruction |
| *AjELMO1*-mut | AATTAAGCAACAGAGACTGAACTACC TCTCTGTTGCTTAATTAATTCCTGAA | Luciferase vector contruction |
| cell division cycle 42-wt | CCCTCGAGTAACGAAGAGAATGAACT GGGTTTAAACTTCAAAAGAGGATGGATA | Luciferase vector contruction |
| cell division cycle 42-mut | TTCTTAGAGTATTGGATGACAAACTT ATTGGCTCAAAGGAAAAGTTTGTCAT | Luciferase vector contruction |
| beclin-1 | AGAAAAGAAATAGCGAAG GAAGACGACCTAGACGAA | Real-time PCR |
| *AjELMO1* | AGAAAGTAGTCCGTGCCC GTAGTCGTCGGCAAACAA | Real-time PCR |
| interleukin-1 receptor-associated kinase | TTTAGGGAAGGGAGGAAG TGGCAGATACGGGAGAGT | Real-time PCR |
| cell division cycle 42 | AGTAGTTGGAGACGGAGC ATTTTCAAACGAGGATGG | Real-time PCR |
| MS2-GFP-*AjELMO1* | CCAAGCTTGACCAAGACCCTCTGTGA CGGAATTCAGAAGTATGTTGATGCCG | MS2-RIP |
| MS2-GFP-*AjELMO1*-mut | AATTAAGCAACAGAGACTGAACTACC TCTCTGTTGCTTAATTAATTCCTGAA | MS2-RIP |

The underline letters represent the restriction enzyme sites.

phagocytosis using the miRanda program. It was found that these four genes (cell division cycle 42, beclin-1, interleukin-1 receptor-associated kinase, and *AjELMO1*) have potential binding sites for miR-2008 (Supplementary Table 2, in Supplementary Information). Then, we detected the results of the abundance of the four genes post miR-2008M treatment in vivo, and found that the

mRNA abundance levels of *AjELMO1* and cell division cycle 42 were significantly affected by the miR-2008M, but the other two genes did not change (Supplementary Fig 3, in Supplementary Information). We further performed the luciferase reporter assay to verify whether miR-2008 could bind to *AjELMO1* or cell division cycle 42. Based on the predicted binding sites between

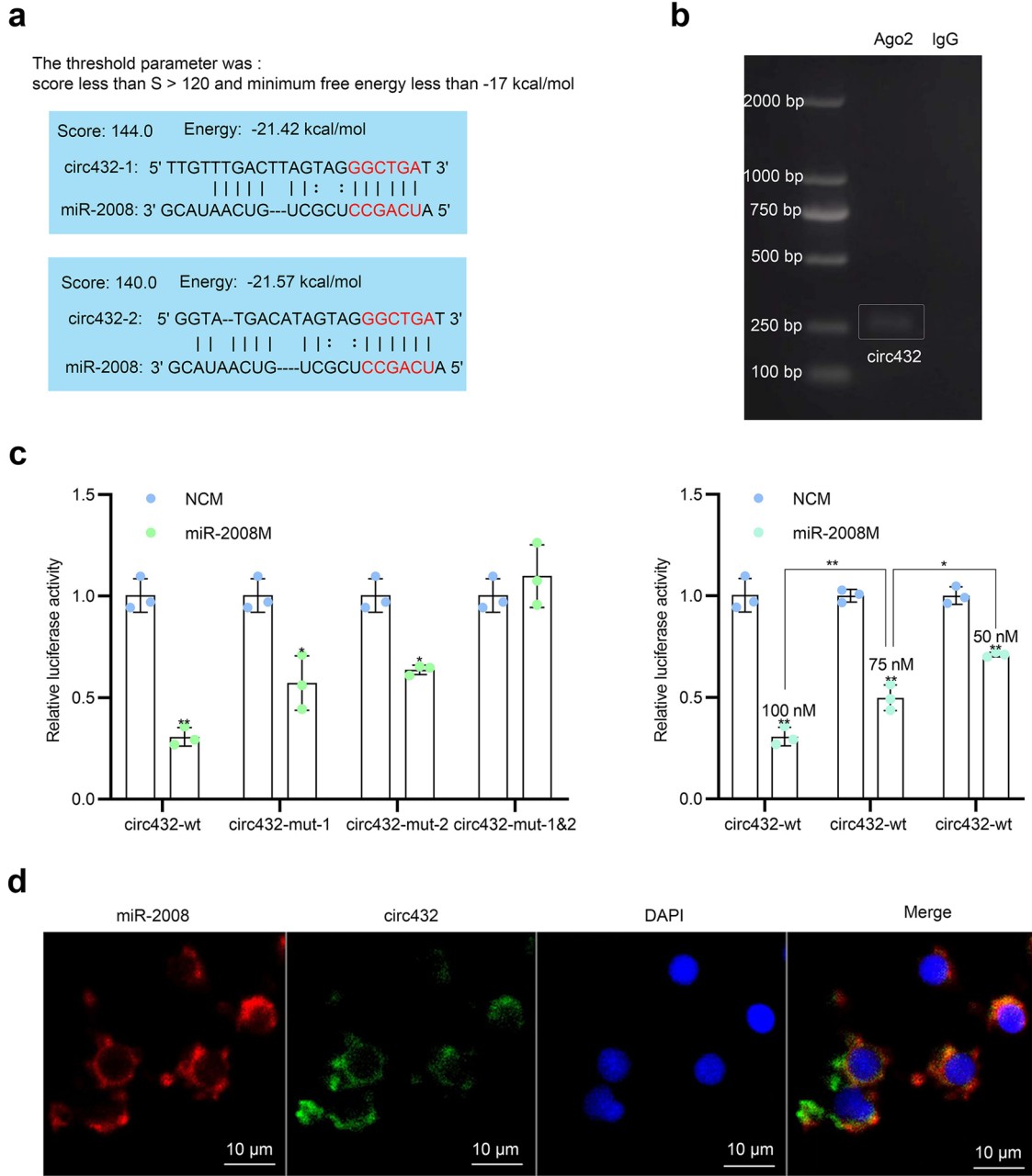

**Fig. 3 circ432 functions as a miRNA sponge of miR-2008. a** The combination of circ432 and miR-2008 was predicted by miRanda. **b** The divergent primers of circ432 of AGO2-RIP were validated by RT-PCR. **c** The relative luciferase activities were detected after co-transfection with circ432-wt or circ432-mut and modified miR-2008 mimics or NCM, and the circ432 relative luciferase activity was detected at different concentration gradients of miR-2008 mimics, including 50, 75, and 100 nM. **d** That circ432 can adsorb miR-2008 was detected by RNA FISH. Scale bar, 10 μm. All data represented the mean ± SD from three independent triplicated experiments. *$P < 0.05$, **$P < 0.01$.

miR-2008 and *AjELMO1*, and cell division cycle 42 (Fig. 5a), we constructed the wild-type (*AjELMO1*-wt, cell division cycle 42-wt) or mutant (*AjELMO1*-mut, cell division cycle 42-mut) luciferase plasmids containing miR-2008 binding sites into the psiCHECK-2 vector (Table 1). After co-transfected Hela cells with luciferase plasmids and miR-2008M for 48 h, we detected that miR-2008M could inhibit the luciferase activity of the *AjELMO1*-wt luciferase plasmid, but had no effect on the *AjELMO1*-mut, cell division cycle 42-wt, and cell division cycle 42-mut plasmids (Fig. 5b, c). In addition, we further detected that miR-2008 mimics inhibited the luciferase activity of *AjELMO1* in a dose-dependent manner (Fig. 5c). Next, to confirm the direct interaction between *AjELMO1* and miR-2008, we constructed

GFP-MS2 (pcGFP-MS2), GFP-MS2-*AjELMO1* (pcGFP-MS2-*AjELMO1*), and GFP-MS2-*AjELMO1*-mut (pcGFP-MS2-*AjELMO1*-mut) plasmids through the pcDNA3-24 × MS2 vector. After co-transfection into Hela cells with miR-2008 mimics and performed with MS2-RIP assay, we detected that pcMS2-GFP-*AjELMO1* could effectively enrich miR-2008, while these results were not shown in the pcMS2-GFP-*AjELMO1*-mut-transfected cells by qPCR (Fig. 5d). These results indicated that *AjELMO1* could function as a potential target gene of miR-2008 for further research. Additionally, we further transfected miR-2008M and miR-2008I into sea cucumbers to examine whether miR-2008 was involved in the regulation of *AjELMO1* abundance. Through in vivo and in vitro interference experiments, the results of the

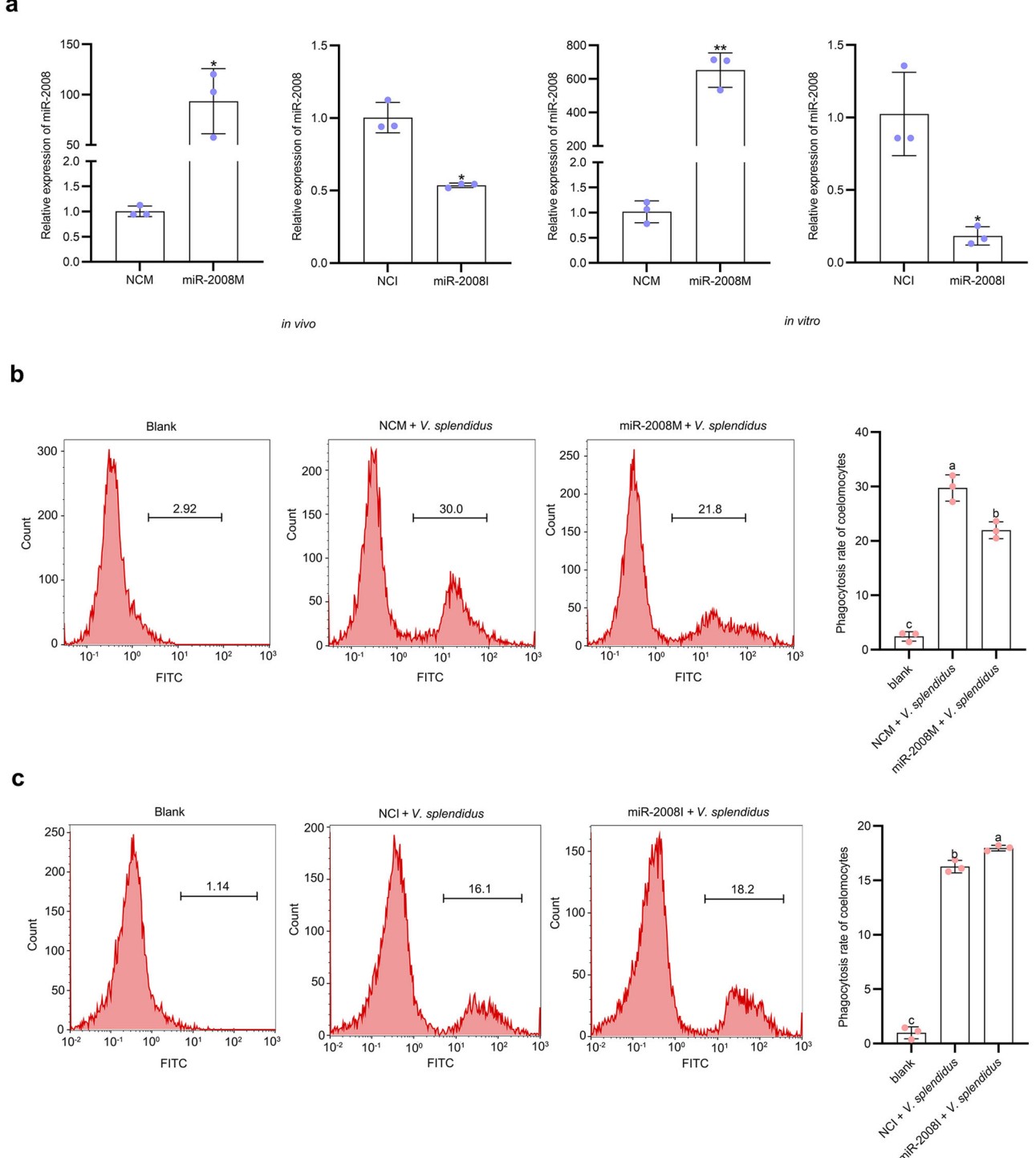

**Fig. 4 Inhibitory effect of miR-2008 on phagocytosis of coelomocytes. a** The relative miR-2008 abundance level after transfection with modified miR-2008 mimics and inhibitor in vivo and in vitro was detected by qPCR. **b, c** The phagocytosis ratio of coelomocytes in modified miR-2008 mimics/inhibitor under the induction of *V. splendidus* was detected by flow cytometry. All data represented the mean ± SD from three independent triplicated experiments. *$P < 0.05$, **$P < 0.01$. Different letters above each bar indicate significant differences: $P < 0.05$, whereas bars with the same letter indicates non-significant differences.

qPCR analysis showed that transfection of miR-2008M in coelomocytes inhibited the mRNA level of *AjELMO1*, while transfection of miR-2008 inhibitor significantly enhanced the mRNA abundance level of *AjELMO1* (Fig. 5e, f). In line with this, the protein level of AjELMO1 also displayed similar results to its mRNA abundance level under the same conditions (Fig. 5e, f). To explore whether miR-2008 mediates *V. splendidus*-induced coelomocytes phagocytosis through *AjELMO1*, we transfected si-

*AjELMO1* into sea cucumber individuals and treated them with miR-2008M for 48 h. The coelomocytes transfected with si-NC and miR-2008M were used as controls. Flow cytometry analysis showed that the phagocytic ratio in coelomocytes of si-*AjELMO1* + miR-2008M + *V. splendidus*-treated sea cucumbers was significantly lower than that in si-*AjELMO1* + NCM + *V. splendidus*-treated sea cucumbers (Fig. 5g). Overall, these results confirmed that AjELMO1 could directly bind to miR-2008 and

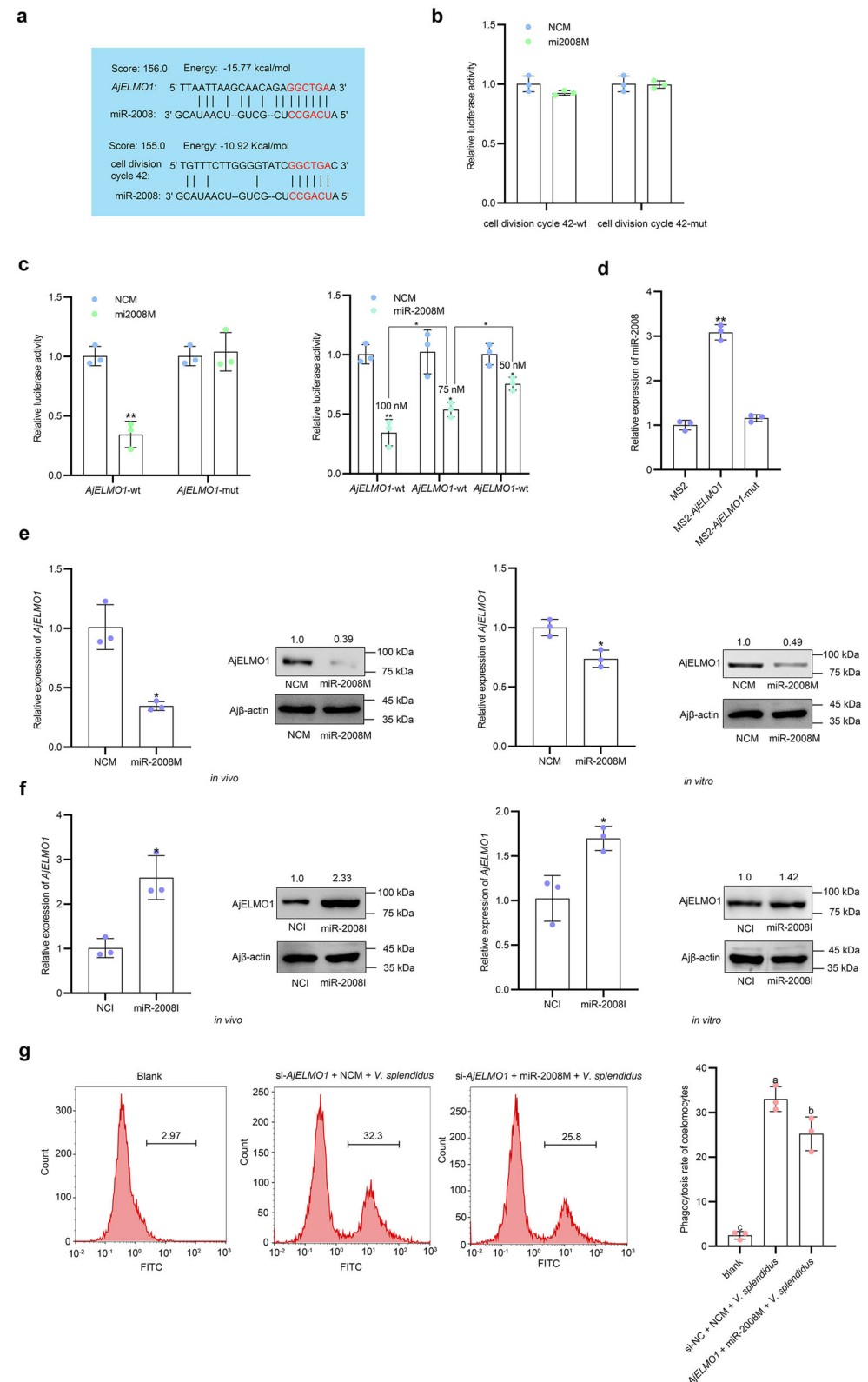

miR-2008 inhibited phagocytosis of coelomocytes through its target AjELMO1.

**AjELMO1 enhances the phagocytic capacity induced by *V. splendidus*.** In previous studies, it had been found that *ELMO1* or its homologous genes mediated the phagocytosis of cells of multiple species, such as mammalian fibroblasts[14] and mouse intestinal epithelial cells[15]. However, the involvement of ELMO1-mediated immune response in sea cucumbers remained unclear. To clarify the immune function of AjELMO1, we cloned its cDNA full length with 4093 bp, including a 5′-untranslated region (UTR) of 139 bp, a 3′-UTR of 1791 bp, and an ORF of 2163 bp (NCBI Accession no. OP218937) and the predicted molecular mass of the deduced aa sequences of AjELMO1 was 81.93 kDa

**Fig. 5 miR-2008 inhibits the *V. splendidus*-induced coelomocytes phagocytosis via targeting AjELMO1. a** The combination of miR-2008 and *AjELMO1*, cell division cycle 42 was predicted by the miRanda program. **b** The relative luciferase activities were detected in Hela cells after co-transfection with cell division cycle 42-wt or cell division cycle 42-mut plasmids and modified miR-2008 mimics or NCM. **c** The relative luciferase activities were detected after co-transfection with *AjELMO1*-wt or *AjELMO1*-mut plasmids and modified miR-2008 mimics or NCM, and the *AjELMO1* relative luciferase activity was detected at different concentration gradients of miR-2008 mimics, including 50, 75, and 100 nM. **d** Hela cells were co-transfected with pcGFP-MS2, pcGFP-MS2-*AjELMO1*, or pcGFP-MS2-*AjELMO1*-mut plasmids and miR-2008 mimics for 48 h, and then incubated with anti-GFP antibody and further detect the enrichment of miR-2008 by qPCR. **e, f** The relative abundance level of AjELMO1 after transfection with modified miR-2008 mimics and inhibitor in vivo and in vitro was detected by qPCR and western blotting. **g** The phagocytosis ratio of coelomocytes in *AjELMO1* siRNA transfection and modified miR-2008 mimics under the induction of *V. splendidus* was detected by flow cytometry. All data represented the mean ± SD from three independent triplicated experiments. *$P < 0.05$, **$P < 0.01$. Different letters above each bar indicate significant differences: $P < 0.05$, whereas bars with the same letter indicates non-significant differences.

(Supplementary Fig 4, in Supplementary Information). SMART analysis revealed that the C-terminus of the sequence had a conserved Elmo domain, which also existed in many eukaryotic proteins (Supplementary Fig 5, in Supplementary Information). The phylogenetic tree (Supplementary Fig 6, in Supplementary Information) and multi-sequence comparison (Supplementary Fig 7, in Supplementary Information) showed that AjELMO1 had high structural conservation and sequence identity with other invertebrates and vertebrates ELMO1s. Moreover, qPCR showed that *AjELMO1* abundance in coelomocytes of *V. splendidus*-challenged sea cucumbers was significantly upregulated at the first 24 h, reaching the peak levels at 24 h, and then decreased to the normal levels at 48 h post-infection compared to the control group. A similar upregulated abundance profile was also detected in LPS-exposed primary coelomocytes. *AjELMO1* rapidly reached the peak abundance levels at 6 h. Subsequently, the abundance of *AjELMO1* decreased continuously and returned to normal levels at 24 h post-treatment (Fig. 6a). Furthermore, we designed a siRNA specifically targeting *AjELMO1* to explore whether it mediates *V. splendidus*-induced coelomocyte phagocytosis in sea cucumbers (Table 1). Through in vivo and in vitro interference experiments, qPCR analysis results showed that the mRNA abundance level of *AjELMO1* was significantly reduced at 48 h post si-*AjELMO1* treatment. In line with this, AjELMO1 protein level was also much decreased under the same conditions (Fig. 6b). Upon successfully knockdown of *AjELMO1* in vivo, we transfected si-*AjELMO1* into sea cucumber individuals for 48 h and then injected FITC-labeled *V. splendidus* for 2 h to analyze the changes of phagocytosis of coelomocytes by flow cytometry. The results analysis showed that the phagocytic ratio in coelomocytes of si-*AjELMO1* + *V. splendidus*-treated sea cucumbers was significantly lower than that in si-NC + *V. splendidus*-treated sea cucumbers (Fig. 6c), which indicated that AjELMO1 positively regulated the phagocytosis of *V. splendidus* by sea cucumber coelomocytes.

**circ432 mediates phagocytosis via miR-2008-AjELMO1 axis.** To prove that circ432 was a ceRNA of miR-2008 and then regulated the expression of AjELMO1, the abundance of AjELMO1 was assayed in the circ432 silencing condition. The results of the qPCR analysis showed that the mRNA abundance levels of *AjELMO1* were significantly decreased after interfering with circ432, but the abundance level of miR-2008 was not affected. Western blotting also showed si-circ432 could significantly decrease the protein expression level of AjELMO1 than si-NC (Fig. 7a). Afterward, si-circ432 and miR-2008I were co-transfected into sea cucumber individuals. The results of the qPCR analysis showed that the mRNA and protein expression levels of AjELMO1 were significantly increased at 48 h after transfection with si-circ432 and miR-2008I (Fig. 7b). Moreover, flow cytometry analysis showed that the phagocytic ratio in

coelomocytes of si-circ432 + miR-2008I + *V. splendidus*-treated sea cucumbers was significantly higher than that in si-NC + NCI + *V. splendidus*-treated sea cucumbers (Fig. 7c). In conclusion, we verified that circ432 functions as a sponge of miR-2008 enhances the abundance of AjELMO1 and promotes the coelomocytes phagocytosis induced by *V. splendidus* in sea cucumbers.

## Discussion

Phagocytosis is the most primitive and initial immune defense system of eukaryotes. In the face of the invasion and infection of exogenous pathogenic microorganisms, the immune response is initiated to realize the encapsulation and removal of pathogenic substances, to ensure the normal physiological and biochemical reactions of the body and the stable response of the internal circulation[5]. As an echinoderm, coelomocytes are the main immune cells of sea cucumbers, which can eliminate microbial pathogens by phagocytosis[41]. ELMO1 was originally identified as a mammalian homolog of *C. elegans* Ced-12, which is necessary for cell migration and phagocytosis of cells[10]. In this study, we cloned the full length of *AjELMO1* from *A. japonicus*, and *AjELMO1* has a conserved ELMO domain at the C-terminus. In addition, phylogenetic trees and multiple sequence comparisons indicated that *AjELMO1* shares a high degree of structural conservation and sequence identity with other invertebrates and vertebrates *ELMO1*. Furthermore, we found that the abundance of *AjELMO1* was increased in *V. splendidus*-challenged sea cucumbers and LPS-stimulated primary coelomocytes, and the phagocytosis activity of coelomocytes was significantly decreased under the induction of *V. splendidus* after silencing *AjELMO1*. These results showed that AjELMO1 plays an important regulatory role in response to pathogen infection.

In recent decades, the research on the regulatory network mechanism of mammalian miRNAs has become increasingly complete and clear, and the research on invertebrate miRNAs is also deepening. For example, in mud crab *Scylla paramamosain*, the differentially expressed miRNAs are mainly related to changes in immune response (including phagocytosis, melanization, and apoptosis) of blood cells[42]; miR-965 can promote the phagocytosis of shrimp *Marsupenaeus japonicus* against viral infection by targeting ATG5 (autophagy-related 5) gene of shrimp[43]; miR-133 promotes *V. splendidus* to participate in TLR cascade regulation of sea cucumber *Apostichopus japonicus* phagocyte through AjIRAK-1 targeting[34]. Commonly, most of the identified miRNA target sites are located in the 3' untranslated region (UTR) of mRNA. The reduced efficacy of the 5' UTR and ORF sites is attributed to the replacement of the miRNA silencing complex in the 5' UTR by the scanning mechanism because it passes from the cap to the start codon and enters the coding region through translocation ribosomes[44]. Nevertheless, some studies have proved that many ORF sites are preferentially conserved[45–47].

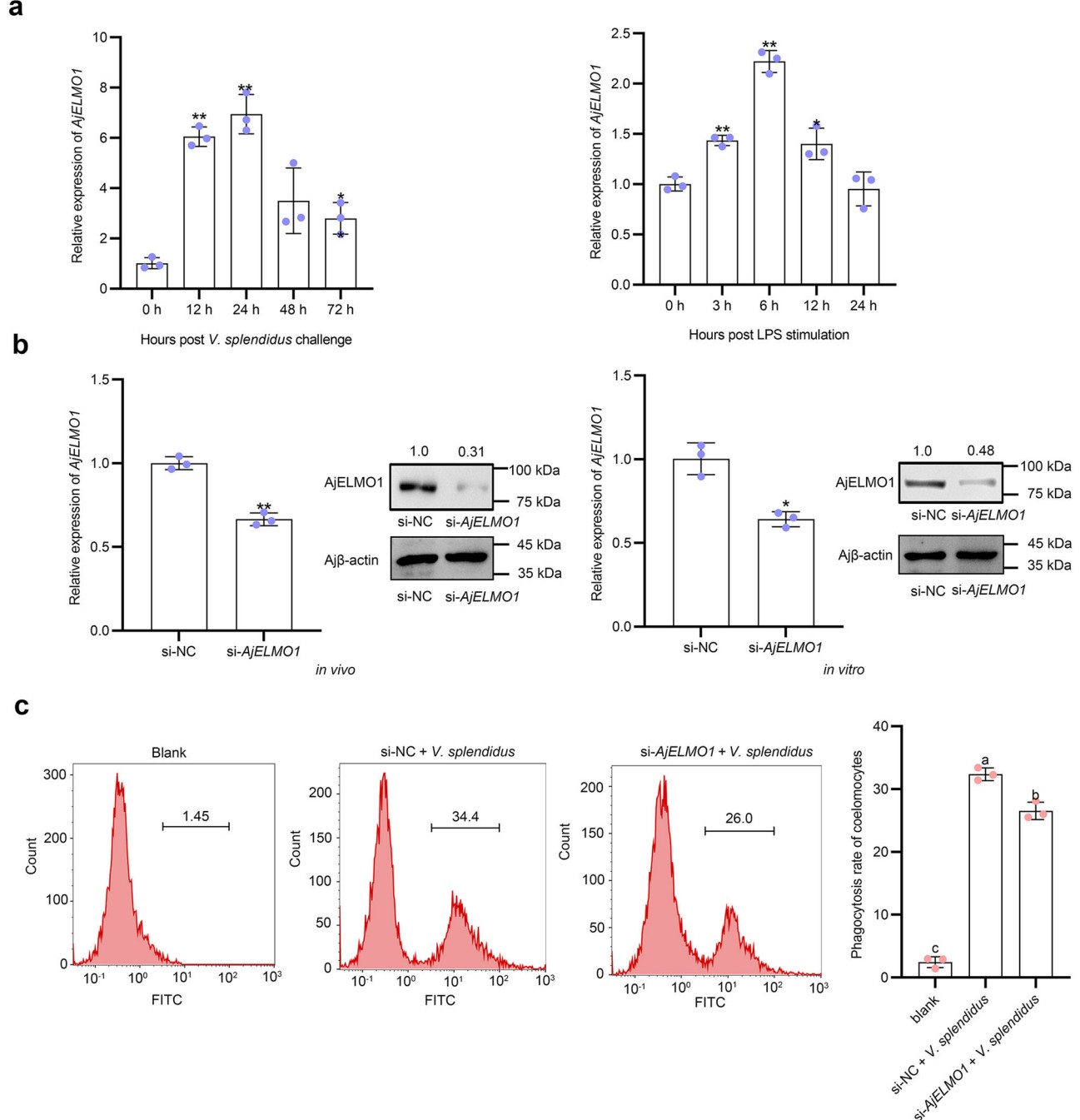

**Fig. 6 AjELMO1 promotes phagocytosis of responses upon *V. splendidus* infection. a** The time-course abundance patterns of *AjELMO1* in the *V. splendidus*-challenged *A. japonicus* and the LPS-exposed primary cultured cell as measured by qPCR. **b** The relative abundance level of *AjELMO1* after *AjELMO1* siRNA transfection in vivo and in vitro was detected by qPCR and western blotting. **c** The phagocytosis ratio of coelomocytes in *AjELMO1* siRNA transfection under the induction of *V. splendidus* was detected by flow cytometry. All data represented the mean ± SD from three independent triplicated experiments. *$P < 0.05$, **$P < 0.01$. Different letters above each bar indicate significant differences: $P < 0.05$, whereas bars with the same letter indicates non-significant differences.

Reporter assays have also confirmed that sites in 5′ UTR and ORFs could mediate repression[48–50]. In this study, the miRanda prediction and luciferase reporter assay indicated that the ORF of *AjELMO1* has a direct binding site for miR-2008. The mRNA level of *AjELMO1* was downregulated after miR-2008 was over-expressed. When miR-2008 was knocked down, the mRNA level of *AjELMO1* was significantly upregulated. The results showed that the mRNA of *AjELMO1* was mainly targeted by miR-2008 through ORF sites. This indicates that the role of miR-2008 in sea

cucumbers has not been evaluated, which has previously pro-moted ROS production by regulating the 3′ UTR of AjBHMT[39], and emphasizes the potential importance of ORF sites in understanding miRNA biology.

The ceRNA hypothesis indicates that RNA transcripts including mRNA, lncRNA, pseudogenes, and circRNAs can compete to disturb and regulate miRNA abundance and share the miRNA response elements (MREs), thereby constructing new and complex post-transcriptional regulatory networks and mechanisms[51].

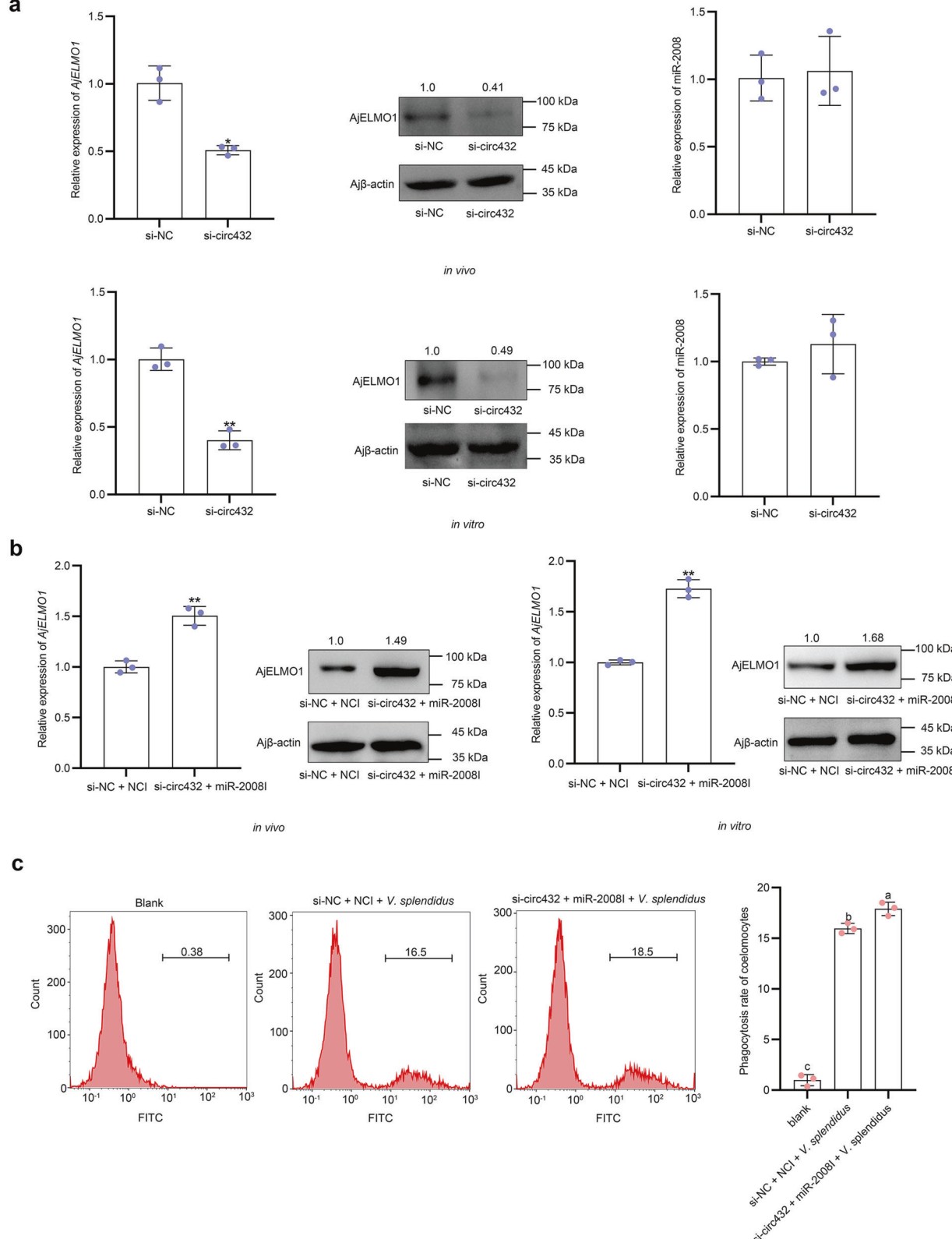

**Fig. 7 circ432 as a sponge for miR-2008 enhances phagocytosis of AjELMO1. a** The relative abundance level of AjELMO1 and miR-2008 after circ432 siRNA transfection in vivo and in vitro was detected by qPCR and western blotting. **b** The relative abundance level of AjELMO1 after circ432 siRNA transfection and modified miR-2008 inhibitor in vivo and in vitro was detected by qPCR and western blotting. **c** The phagocytosis ratio of coelomocytes in circ432 siRNA transfection and modified miR-2008 inhibitor under the induction of *V. splendidus* was detected by flow cytometry. All data represented the mean ± SD from three independent triplicated experiments. *$P < 0.05$, **$P < 0.01$. Different letters above each bar indicate significant differences: $P < 0.05$, whereas bars with the same letter indicates non-significant differences.

Among them, many studies have shown that circRNAs can be used as miRNA sponges to regulate the abundance of mRNA target genes[23–26]. In this study, the miRanda prediction, luciferase reporter assay, and RIP assay indicated that circ432 had a high binding capacity with miR-2008. At the cellular level, RNA FISH revealed that circ432 was present in the cytoplasm and co-localized with miR-2008. These results showed that circ432 might function as a sponge of miR-2008. In addition, through in vivo and in vitro interference experiments, qPCR and western blotting analysis results showed that knocking out circ432 significantly reduced the abundance level of AjELMO1, but did not affect the abundance level of miR-2008. These results confirm that circ432 has a typical ceRNA mechanism, it can act as a sponge to adsorb miR-2008, indirectly enhance the abundance of AjELMO1, and promote the antibacterial immune response of sea cucumbers. Additionally, when examining the tissue specificity of circ432, circ432 was also expressed in other tissues (intestine, muscle, etc.), suggesting that circ432-miR-2008-AjELMO1 may also be present in other tissues. However, the RNA network has a multi-interactive, multi-layered, multi-dimensional, and multi-directional network architecture that can coordinate cellular responses. Whether circ432 communicates with other miRNAs/mRNAs/RBPs to form an RNA regulatory network remains to be further confirmed.

In summary, we report a circ432-miR-2008-AjELMO1 axis interaction network that mediates *V. splendidus*-induced coelomocytes phagocytosis in sea cucumbers (Fig. 8). miR-2008 can reduce the abundance of AjELMO1 and inhibit AjELMO1-mediated phagocytosis of coelomocytes, which may help *V. splendidus* to evade the host antimicrobial response. We further demonstrated that circ432 as an endogenous sponge RNA interacted with miR-2008 and promoted the abundance of AjELMO1, thereby enhancing the phagocytosis of sea cucumber coelomocytes induced by *V. splendidus*. The regulatory network involving the circRNA-miRNA-mRNA axis might provide ideas for the targeted treatment of SUS.

## Methods

**Ethics statement**. The sea cucumbers *A. japonicus* used in this work were commercially cultured animals, and all experiments were conducted in accordance with the recommendations in the Guide for the Care and Use of Laboratory Animals of the National Institutes of Health. The study protocol was approved by the Experimental Animal Ethics Committee of Ningbo University, China (No. NBU-ES-2021-11180).

**Sample and challenge**. Healthy adult sea cucumber *A. japonicus* ($120 \pm 5$ g) were purchased from Xinyulong Aquaculture Company (Dalian, China). The species, size, breeding, and infection conditions of the sea cucumbers we used were all carried out by Zhao et al.[29]. Before the start of the experiment, the sea cucumbers were temporarily cultured in seawater (salinity 28 PSU, temperature 16 °C) for 3 days. In the challenge experiment, 40 sea cucumbers were evenly divided into two tanks and immersed in high-density live *V. splendidus* with a final concentration of $10^7$ CFU mL$^{-1}$. The same number of individuals in the other two tanks served as controls. The coelomic fluids were collected from the individuals in each tank and centrifuged at $800 \times g$, 4 °C for 5 min to harvest coelomocytes at 0, 12, 24, 48, and 72 h. For spatial expression analysis, five tissues including coelomocytes, muscle, tentacle, respiratory trees, and intestine were collected from the untreated group. These tissues were homogenized into powder in liquid nitrogen using a mortar and a pestle. The samples were then stored at −80 °C for RNA extraction and cDNA synthesis.

**Primary coelomocytes culture and LPS exposure**. The cells are collected by centrifugation at $800 \times g$ at 4 °C for 10 min and mixed with an equal volume of anticoagulant solution (0.02 M EGTA, 0.48 M NaCl, 0.019 M KCl, 0.068 M Tris −HCl, pH 7.6). Then the washed cells were re-suspended in the Leibovitz's L-15 (Meilunbio, Dalian, China) containing penicillin (100 U/mL$^{-1}$), streptomycin sulfate (100 mg mL$^{-1}$), and NaCl (0.39 M) at a final concentration of $10^6$ cells mL$^{-1}$. The equalized sample of 500 μL cell suspension was allocated to a 24-well culture plate and incubated at 16 °C for 24 h before LPS treatment (Solarbio, Beijing, China). For LPS stimulation, the primary cultured coelomocytes were exposed to LPS at the final concentration of 1 mg mL$^{-1}$ for 0, 3, 6, 12, and 24 h. We performed three replicates for each of the experimental groups as well as the control group.

**RNA isolation and real-time quantitative PCR**. The cytoplasmic and nuclear RNA of coelomocytes was isolated and purified using the Cytoplasmic & Nuclear RNA Purification Kit based on the manufacturer's instructions (Norgen Biotek Corp, Ontario, Canada). Total RNA was isolated using RNAiso Plus (TaKaRa, Ootsu, Japan) and the cDNA was synthesized using miScript II RT Kit (QIAGEN, Frankfurt, German) and PrimeScript RT reagent Kit with gDNA Eraser (TaKaRa, Ootsu, Japan). RT-qPCR assay targeting *AjELMO1*, circ432, and Ajβ-actin that have been designed, optimized, and validated in strict compliance with the MIQE guidelines[52]. Detailed information is shown in the Supplementary materials (Supplementary Note 1, in Supplementary Information). For validation of the expression stability of the housekeeping gene in healthy sea cucumber tissues and in sea cucumber coelomocytes post-*V. splendidus* challenge, we selected ten candidate reference genes based on the frequently used housekeeping genes and genes that showed stable expression in our acquired RNA-seq transcriptome data[40], including Ajβ-actin, 40 S ribosomal protein S9 (RPS9), RPS18, glyceraldehyde 3-phosphate dehydrogenase (GAPD), NADH dehydrogenase (NADH), α-Tubulin (TUBA), β-Tubulin (TUBB), elongation factor-1 (EF1α), NADH dehydrogenase [ubiquinone] 1α subcomplex subunit 13 (NDUFA13), and 60 S ribosomal protein L18a (RPL18A). The expression stability of these ten primers were analyzed by four algorithms, including NormFinder (version 0.953)[53], geNorm (qbase + , version 3.1)[54], BestKeeper (version 1)[55], and RefFinder (https://blooge.cn/RefFinder/?type=reference) in *V. splendidus*-challenged coelomocytes of *A. japonicus* at five different time points (0, 12, 24, 48, and 72 h) and five different tissues, including coelomocytes, intestines, muscles, tentacles, and respiratory trees with three biological replicates. All the primer information except for verifying the stable expression of housekeeping genes is shown in Table 1, and the other genes are shown in the Supplementary materials (Supplementary Note 2, in Supplementary Information). RNU6B and a stably expressed housekeeping gene screened above served as the internal control to standardize the miRNA or the target genes for quantification. The final volume of each reaction was 20 μL, including 2 μL cDNA, 1 μL each primer (10 μM), 6 μL RNase-free water, and 10 μL SYBR Green PCR Master Mix (TaKaRa, Ootsu, Japan). The amplification profile was as follows: denaturation at 94 °C for 5 min, followed by 40 cycles of 94 °C for 15 s, and 60 °C for 35 s. RT-qPCR reactions were performed using 7500 real-time PCR detection system (Applied Biosystems, Singapore) and LightCycler480 II instruments (Roche, Basel, Basel Stadt, Switzerland). The abundance level of candidate genes was analyzed by the $2^{-\Delta\Delta CT}$ method[31].

**RNase R treatment**. The RNAs (2 μg) extracted from coelomocytes were treated with 1 μL RNase R (10 U/μg, Geneseed, Guangzhou, China) and incubated at 37 °C for 15 min. After the reaction, the treated RNAs were subjected to reverse transcription, and the final product cDNA was used for RT-PCR to detect divergent primers and convergent primers of circRNA loop formation. The sequence information is shown in Table 1.

**Prediction of the complementarity among circ432, miR-2008, and AjELMO1**. In our previous studies, the transcriptome data of circRNAs (BioProject ID: PRJNA512578)[29], 7 miRNAs[32–38], and mRNAs[40] (BioProject ID: SRA080354) in coelomocytes of healthy and SUS-diseased sea cucumbers were identified by the high-throughput sequencing technology. Then, we predicted whether circ432 could be bound with these seven miRNAs by the miRanda program. The screening threshold: fraction less than S > 120 (single residue fraction) and minimum free energy less than −17 kcal/mol (Table S1). Next, we further screened out genes that might be related to phagocytosis in the transcriptome of *A. japonicus*[40]. After filtering some genes without complete sequence annotation, the binding sites of miR-2008 and genes potentially related to phagocytosis were predicted using the miRanda program.

**Dual-luciferase reporter vector detection**. The sequences of circ432 and *AjELMO1* containing miR-2008 binding sites and the mutated sequence (Table 1) were cloned into the plasmid of the psiCHECK-2 vector (Promega, Beijing, China) and verified by sequencing (Sangon Biotechnology, Shanghai, China). The single-site and multi-site mutations of circ432 and *AjELMO1* sequence were performed using the point mutagenesis kits (Transgen, Beijing, China) according to the manufacturer's instruction. Then, 100 μL Hela cells ($2 \times 10^4$ cells/well) were inoculated into a 96-well plate, luciferase vector containing miRNA binding sites or mutation sites (200 ng), Renilla luciferase vector (20 ng), and miRNA mimics (100 ng) were co-transfected into Hela cells for transfection. At 48 h post-transfection, the cells were lysed to obtain reporter gene activity. All the luciferase activity values were obtained for the renilla luciferase control. The transfection of each construct was performed in triplicate in each assay. For each experiment, the ratio of renilla luciferase readings to firefly luciferase readings was taken, and the average was performed by three replicates.

**RNA-fluorescence in situ hybridization (RNA FISH)**. Coelomocytes were collected and cultured as described above. Then, the coelomocytes ($10^6$ cells mL$^{-1}$)

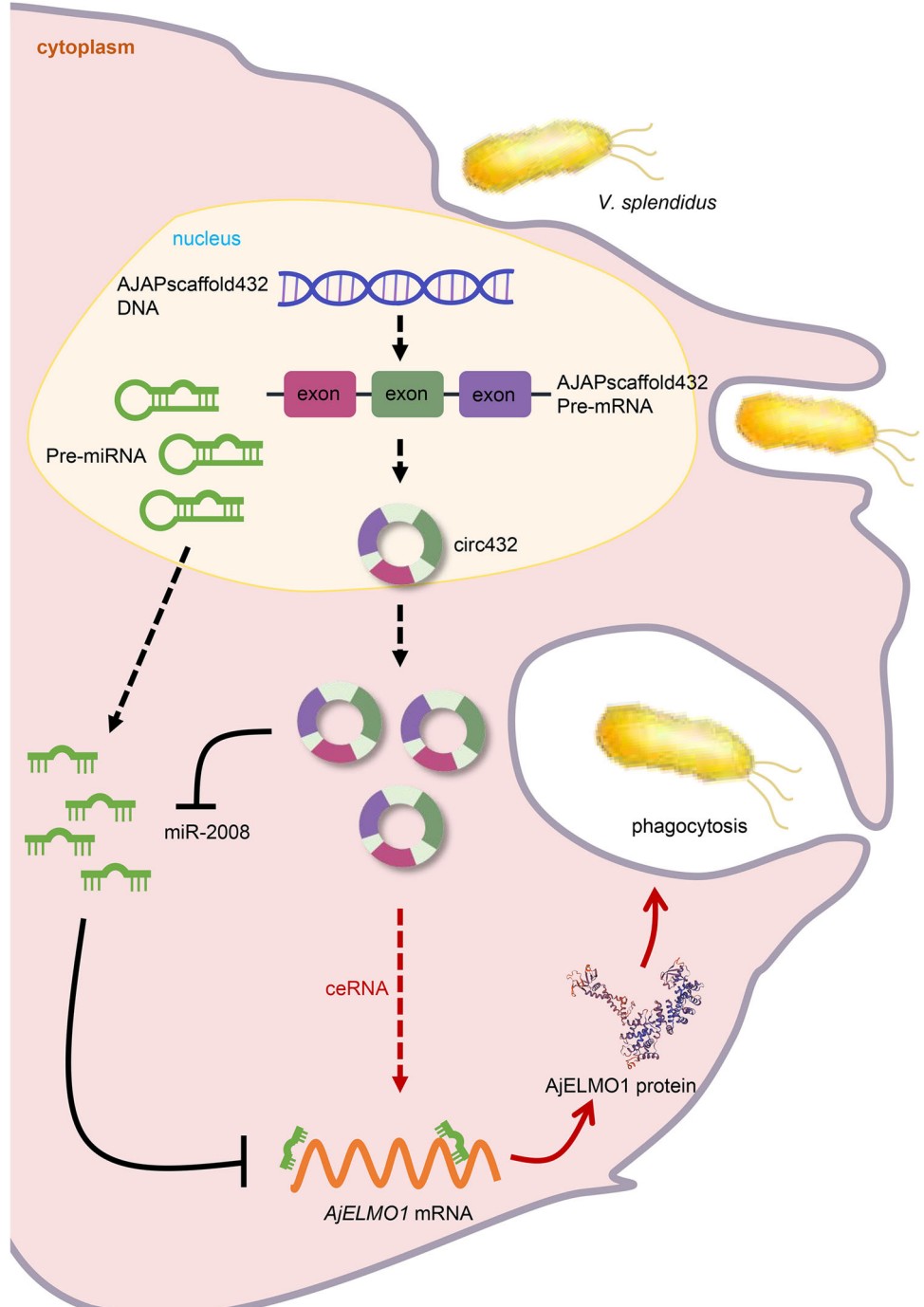

**Fig. 8 The schematic diagram shows the mechanism underlying circ432 as a ceRNA for miR-2008 to regulate *V. splendidus*-induced coelomocytes phagocytosis by targeting AjELMO1.** miR-2008 targets AjELMO1 and represses AjELMO1-mediated coelomocytes phagocytosis. circ432 acts as a molecular sponge regulating miR-2008 to enhance AjELMO1 abundance to maintain the phagocytosis balance.

were treated with LPS (1 mg mL$^{-1}$). After LPS treatment for 24 h, the coelomocytes were washed twice with phosphate buffered saline (PBS) and fixed in 4% for-maldehyde for 15 min. The fixed cells were transfused with Triton X-100 and dehydrated through a series of increasing ethanol concentrations. The cells were incubated with a 50 nmol probe and cultured at 73 °C for 5 min. The cells were hybridized at 37 °C for 14 h, washed, and dehydrated. After adding DAPI working solution, the cells were scanned and imaged. RNA FISH probes (FITC-labeled circ432 and Cy3-labeled miR-2008) were designed and synthesized by Bio-technology Corporation (Shanghai, China).

**Cloning and sequence analysis of the AjELMO1 protein.** The partial ELMO1 cDNA sequence was extracted from our completed *A. japonicus* transcriptome data[30]. Then, the 3′, 5′-complete RACE kit (TaKaRa, Ootsu, Japan) was used to determine the full-length cDNA sequence of *AjELMO1*, and the specific primers of *AjELMO1* were designed according to the acquired sequence in Table 1. The obtained PCR products were directly sequenced at Sangon Biotechnology (Shanghai, China). BLAST algorithm (http://www.ncbi.nlm.nih.gov/blast) was used to analyze the *AjELMO1* cDNA sequence. The amino acid (aa) sequence and domain characteristics were analyzed by an expert protein analysis system (http://www.expasy.org/) and the SMART program (http://smart.embl-heidelberg.de/). The percentage of sequence similarity between *AjELMO1* and *ELMO1* from other organisms was calculated using the Identity and Similarity Analysis program (http://www.biosoft.net/sms/index.html). In the MEGA7.0 software package, ClustalW aligned protein sequences, the NJ distance method estimated pairwise distances, and bootstrapped support values of 10,000 replicates to construct phylogenetic trees.

**Prokaryotic expression and preparation of AjELMO1 polyclonal antibody**. The open reading frame (ORF) of *AjELMO1* was amplified using the specific primers listed in Table 1. The amplified PCR products were digested by the restriction enzymes (*Xho*I and *Hin*dIII) and ligated into the pET-28a (+) expression vector (Novagen, Madison, WI, USA). Then, the recombinant plasmid was transformed into *Escherichia coli* Rosetta cells (Novagen, Madison, WI, USA). After induction with 1 mM isopropyl-β-d-thiogalactopyranoside (IPTG), the bacteria were collected and the target protein was purified according to our previous study[35]. The recombinant AjELMO1 protein was purified by using the Ni-NTA sepharose column (Sangon Biotechnology, Shanghai, China). The purified recombinant protein was determined by 12% SDS-PAGE. The concentration of rAjELMO1 was determined by the BCA detection kit (Sangon Biotechnology, Shanghai, China). The AjELMO1 polyclonal antibody was generated by injecting the protein into a 4-week-old mouse. The antiserum was stored at −80 °C for subsequent experiments.

**Western blotting**. The proteins of the control and treated coelomocyte were extracted using the Total Protein Extraction Kit (Sangon, Shanghai, China), and the protein concentration was measured with a BCA Protein Assay Kit (Sangon, Shanghai, China). Approximately 50 μg of the protein was separated by 12% SDS-PAGE and transferred to 0.45 μm PVDF membrane by electrophoresis. After blocking with 5% skimmed milk in TBST (1% Tween-20, 150 mmol L⁻¹ NaCl, and 50 mmol L⁻¹ Tris-HCl) at 37 °C for 2 h, the membranes were incubated with the polyclonal antibody against AjELMO1 (1: 400) or β-actin (1: 2000, Abmart M20027S, Shanghai, China) at 37 °C for 2 h and subsequently incubated with goat-anti-rat IgG (1: 5000, Beyotime D110087-0100, Shanghai, China) at 37 °C for 2 h. After being washed thrice with TBST, the membrane was incubated with an Omega Lum C imaging system (Aplegen, California, USA). Bands were quantitatively analyzed using the ImageJcngr software package, and the gray value was indicated as the statistical analysis of three independent experiments.

**In vitro and in vivo functional analysis of circ432, miR-2008, and *AjELMO1***. circ432 siRNA (si-circ432), *AjELMO1* siRNA (si-*AjELMO1*), miR-2008 mimics/inhibitor (miR-2008M/miR-2008I) and their negative controls (si-NC, NC mimics (NCM), NC inhibitor (NCI) were synthesized by GenePharma Company (Shanghai, China). The detailed sequence information is shown in Table 1. For in vitro transfection experiments, 1 μL si-circ432, si-*AjELMO1*, or miR-2008M/miR-2008I and their negative controls (si-NC, NCM, NCI) were mixed with 1 μL Lipo6000 transfection reagent (Beyotime, Shanghai, China). The mixture was then transfected into 500 μL of primary cultured cells in each well and incubated in the dark for 48 h before use at 16 °C. For the in vivo experiment, 10 μL circ432 siRNA, *AjELMO1* siRNA or miR-2008M/miR-2008I, and negative control were mixed with 10 μL Lipo6000 transfection reagent and 80 μL PBS as the transfection solution. Sea cucumbers (100 ± 10 g) were infected with 100 μL of the above transfection solution. Control and treated coelomocytes were collected for further RNA and protein extraction 48 h after being treated.

**RNA immunoprecipitation assay (RIP)**. RIP experiments were performed by using the Magna RIP RNA-Binding Protein Immunoprecipitation Kit (MilliporeSigma, Burlington, MA, USA) following the manufacturer's protocol. The pre-treated coelomocyte protein extracts were first precleared using the protein A + G agarose beads. Then, control IgG and the Argonaute 2 (AGO2) primary monoclonal antibody (1: 1000, Abmart ab186733, Shanghai, China) were supplemented to the supernatants, and followed by incubation with protein A + G beads at 4 °C for 10 h. Subsequently, the RNA-protein complexes were washed with PBS in a triple and digested with proteinase K. After the last washed in a triple with PBS, the RNA was isolated with Trizol and detected by an agarose gel electrophoresis analysis. MS2-RIP was performed to construct the MS2 and GFP fusion expression plasmids to produce GFP-MS2 fusion protein that can bind to MS2 fragment and further identify with anti-GFP antibody (1: 2000, Abcam ab6556, Shanghai, China). Next, after transfected pcMS2-GFP, pcMS2-GFP-*AjELMO1* and pcMS2-GFP-*AjELMO1*-mut plasmids through the pcDNA3-24 × MS2 vector (Miaoling, Wuhan, China) in Hela cells for 48 h, the pre-treated cell protein extracts were first precleared using the protein A + G agarose beads. Then, the anti-GFP antibody was supplemented with the supernatants, and followed by incubation with protein A + G beads at 4 °C for 10 h. RNA was extracted from the remaining beads and qPCR was used to evaluate the expression levels of miRNAs.

**Cell phagocytosis assay**. Two hyndred microliters of FITC-labeled *V. splendidus* (OD₆₀₀ = 1.0) was injected into the sea cucumber after 48 h of in vivo interference, then the coelomocytes were harvested after 2 h of *V. splendidus* challenge, phagocytic activity was examined by a Flow Cytometer (Thermo Scientific, Madison, WI, USA).

**Statistics and reproducibility**. The data of gene abundance and cell phagocytosis assay were expressed as the relative expression level (mean ± standard deviation, $n = 3$), and then one-way analysis of variance (ANOVA) and multiple Duncan tests was performed to determine the mean difference controls. The $t$-test was used to mark the asterisk between the two groups. One asterisk for $0.05 > P > 0.01$ and two asterisks for $P < 0.01$. The analysis of variance (ANOVA) was used to mark

different "letters" between at least three groups. Different letters above each bar indicate significant differences: $P < 0.05$, whereas bars with the same letter indicates non-significant differences.

**Reporting summary**. Further information on research design is available in the Nature Portfolio Reporting Summary linked to this article.

## Data availability

The dataset generated and analyzed during this study is shown in Supplementary Data 1, and the uncut protein imprinting is shown in Supplementary Fig 8 (Supplementary Information).

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

## Acknowledgements

This work was financially supported by Natural Science Foundation of Zhejiang Province (LY22C190004, LZ19C190001), National Natural Science Foundation of China (31902399, 32073003), Key Project from Science Technology Department of Zhejiang Province (2019R52016), Young Doctor Innovation Research Project of Ningbo Natural Science Foundation (20221J089), Ningbo Natural Science Foundation (2021J113), and the K.C. Wong Magna Fund in Ningbo University.

## Author contributions

X.M.F. performed the experiments, interpreted the data, and wrote the manuscript; M.G. and C.H.L. participated in the experimental design, interpreted the data, and revised the manuscript. J.Q.L. and C.H.L. contributed new reagents and analytic tools. All authors read and approved the final version of the manuscript.

## Competing interests

The authors declare no competing interests.
