## [Peer Review File · Communications Biology]

Reviewers' comments:

Reviewer #1 (Remarks to the Author):

The authors found that 261 differentially expressed circRNAs including circRNA432 (circ432) were identified from skin ulcer syndrome (SUS) diseased sea cucumber *Apostichopus japonicus* by RNA-seq. Moreover, the author also found a novel engulfment and cell motility protein 1 (AjElmo1) which could promote the *V. splendidus*-induced coelomocytes phagocytosis. Through further research, they found that circ432 was involved in regulating pathogen-induced coelomocyte phagocytosis via sponge miR-2008 and promoted the abundance of AjElmo1.

After reading this manuscript carefully, I found that the data in this manuscript is not enough to support its final conclusion, lacking many key experiments. In addition, there are many places in this manuscript that make me puzzled, and the logic of its experimental design is very inappropriate. Moreover, I was surprised that the authors only conducted some simple double luciferase reporter experiments to prove their conclusions. I think that the conclusions reached by only a little single experiment are completely unreliable. Finally, I did not see any *V. splendidus* infection experiments in this manuscript, but the author repeatedly emphasized in the abstract that their research was carried out under the condition of *V. splendidus* infection, which is very contradictory. Also, most of the pictures in this manuscript are ambiguous. So I think this manuscript does not have enough credibility to be published in the current version.

Reviewer #2 (Remarks to the Author):

In this study, Fu et al. identified circ432, an intergenic-type circRNA from *A. japonicus*, involved in regulating pathogen-induced coelomocyte phagocytosis via sponge miR-2008 and promoted the abundance of AjElmo1 in sea cucumbers. It was demonstrated that miR-2008 can reduce the translation of AjElmo1 and inhibit AjElmo1-mediated phagocytosis of coelomocytes, which may help *V. splendidus* to evade the host antimicrobial response. The authors further demonstrated that circ432 as an endogenous sponge RNA interacted with miR-2008, and the depressed coelomocytes' phagocytosis by circ432 silencing was consistent with the decreased abundance of AjElmo1, and could be recovered by miR-2008 inhibitors transfection. This study employed multiple techniques within a combination of *in vivo* and *in vitro* studies to confirm the conclusion. This is a very interesting work on how the circRNA exerts its immune function in a model species that possesses a unique evolutionary status from invertebrate to vertebrate differentiation. Overall, I find that this manuscript is well written and suitable for publication in *Communications Biology*. However, some critical points need to be addressed to fully support the conclusion. My comments are as follows:

1. For all the figures, a higher font size and resolution would be desired.
2. In Fig. 1A, I don't see whether circ432 is composed of exons or introns and sequence information. The author had better point out the composition of circRNA composed of exons or introns and write the sequence of circ432.
3. In Fig. 1E, the scale should be marked in the micrograph.
4. The author should carefully examine and unify the statistical comparison method and the marking of asterisks used in the whole article. For example, a t-test was used to compare the significant difference between two groups, and the ANOVA was performed among at least three groups or various time points.
5. In Fig. 2D, where is the data of the phagocytosis ratio of coelomocytes in the blank group? I did not see the data above the scale on the left?
6. In Fig. 5, Fig. 6, and Fig. 7, please show the experimental design briefly and clearly under each band, and it is best to show the band size and the gray value of the target band.
7. Did you do flow cytometry with cells transfected with circ432 or a construct that has miR-2008 binding sites mutate? Wouldn't this be a more direct test of the effect of circRNA432 on phagocytosis (and potential role in immunity)?
8. siRNA is well known to have off-target effects, while the authors designed two different siRNAs, only one was used for the functional experiments. How the authors confirmed the results could not be due to the off-target effects.
9. In the section of "Materials and methods", please supplement the description of "western

blotting" in the article. Besides, please provide the detail information of all the antibody used in this study.

10. In the section of "Prediction of the complementarity among circ432, miR-2008 and AjElmo1" (line 188), did the transcriptome data of circRNAs, miRNAs, and mRNAs in this article uploaded? Please show the accessible ID of the dataset.

11. The database used in the article must be uploaded and accessible. The access number of circ432 and AjElmo1 sequence should be uploaded.

Reviewer #3 (Remarks to the Author):

This study aimed at unravelling the phagocytic molecular mechanism regulated by AjElmo1 under circ432 control as a ceRNA for miR-2008 in sea cucumber coelomocytes. The authors based this study on their own preliminary data and elsewhere. The research is innovative and the data are original. The study follow a logical workflow and uses several complementary techniques including qPCR, sequencing, flow cytometry, dual luciferase reporter, bioinformatic and protein evaluation. I have 2 concerns about the methodologies:

1- The authors have chosen to use Sybr Green for RT-qPCR. However, there is no indication in the manuscript about validation of the methodology according to the MIQE guidelines (Bustin et al., Clin Chem. 2009 Apr;55(4):611-22. doi: 10.1373/clinchem.2008.112797). For instance, there is no controls to demonstrate no genomic amplification or no specific amplification by the technique. In addition, there is no mention of the validation of housekeeping genes used to normalised the relative quantification.

2- Caelomocytes have capacity to aggregates. This could be a challenge for accurately count the number of cells between treatment and also normalise the number of cells for RNA, protein and cell count. How did the author overcome this challenge?

Dear Reviewers,

Thank you very much for your letter and the comments about our paper “circRNA432 enhances the coelomocyte phagocytosis via regulating the miR-2008-Elmo1 axis in *Vibrio splendidus*-challenged *Apostichopus japonicus* (COMMSBIO-22-2620), which helped improve our paper’s quality. We have considered the comments and corrected them carefully. We also proofread our manuscript very closely for mistakes and grammatical errors. Here we submit a revised manuscript with color-coded as well as detailed responses. We appreciate your warm work earnestly, and hope that the revised manuscript will meet with approval. The following is our response to each comment point by point.

Reviewer 1#

The authors found that 261 differentially expressed circRNAs including circRNA432 (circ432) were identified from skin ulcer syndrome (SUS) diseased sea cucumber *Apostichopus japonicus* by RNA-seq. Moreover, the author also found a novel engulfment and cell motility protein 1(AjElmo1) which could promote the *V. splendidus*-induced coelomocytes phagocytosis. Through further research, they found that circ432 was involved in regulating pathogen-induced coelomocyte phagocytosis via sponge miR-2008 and promoted the abundance of AjElmo1.

Response: Yes, you correctly interpret our major finding in this manuscript.

Comment 1: There are many places in this manuscript that make me puzzled, and the logic of its experimental design is inappropriate.

Response: We are very sorry for our unclear description of the experimental design. According to your professional and sincere comments, we have revised the whole manuscript where the logic is not rigorous and have rearranged the logic of our experimental design for the convenience of readers. In our previous work, 261 differentially expressed circRNAs including circ432 were identified in coelomocytes between healthy and SUS-diseased *A. japonicus* sea cucumbers (Zhao et al., Frontier

in Genetics, 2019, 00603). We selected circ432 for subsequent research due to the following two reasons. Firstly, it was highly expressed in coelomocytes of SUS-diseased sea cucumbers by RNA-seq (up-regulated abundance 8.023-fold). Secondly, according to the ref. Zhao et al., 2019, 71.6% of differentially expressed circRNAs were intergenic-type circRNAs, indicating that most of the sea cucumber circRNAs are much longer, and this is different from other species, such as humans and mice. circ432 was also an intergenic circRNA with 2358 bp in length (**Fig. S1**), which is much shorter than most other identified circRNAs in *A. japonicus*. This is beneficial for the subsequent verification of the circ432 function. Subsequently, we detected whether circ432 is an exact circRNA by the Sanger sequencing, RNase R treatment, and divergent primers (**Fig. 1a, Fig. 1b, Fig. 1c**), and further confirmed that it was significantly induced in coelomocytes with a 7.21-fold and 2.36-fold post *V. splendidus* challenge and LPS stimulation (**Fig. 2b**), suggesting that circ432 plays a vital role in mediating pathogen invasion. Next, given that *V. splendidus* is a facultative intracellular bacterium (Liu et al., 2010, J Invertebr Pathol. 105, 236-242) and phagocytosis is the first process of cellular immunity, we thus detected whether circ432 participates in the process of pathogen-induced cell phagocytosis. More importantly, we indeed demonstrated that circ432 is a positive regulator involved in *V. splendidus*-induced sea cucumber coelomocyte phagocytosis (**Fig. 2d**). Since circRNAs were predominantly located in the cytoplasm and usually function as miRNA sponges (Kulcheski et al., J Biotechnol, 2016, 238: 42-51; Chen et al., Biochem Biophys Res Commun, 2017, 494: 126-132). We detected that circ432 was located in the cytoplasm (**Fig. 1e, Fig. 2d**) and further predicted whether circ432 could be bound with the 7 miRNAs that related to SUS-diseased sea cucumbers in our previous studies by the miRanda program (**Table S1**). Through the prediction analysis with the screening threshold, miR-2008 showed the highest possibility to bind circ432 (**Fig. 3a**), we thus selected circ432 and miR-2008 for follow-up research in this study. We further detected that circ432 functions as a miRNA sponge of miR-2008 by the AGO2-RIP assay, the double luciferase report assay, and the RNA FISH (**Fig. 3b, Fig. 3c, Fig. 3d**). Upon the successful results of the binding of circ432 and miR-2008, we

further explored whether miR-2008 could regulate *V. splendidus*-mediated coelomocytes' phagocytosis in sea cucumbers. Through the flow cytometry analysis, it was found that miR-2008 inhibits *V. splendidus*-induced coelomocytes phagocytosis (**Fig. 4b**). For miR-2008 targets analysis, we previously confirmed that miR-2008 regulates ROS production, and ROS is produced mainly by phagocytosis in echinoderms (Coteur et al., Fish Shellfish Immunol, 2002, 12, 187-200). We thus focused on the *A. japonicus* transcriptome (Zhang et al., Fish Shellfish Immun, 2014, 38, 383-388) to screen for genes that might be related to phagocytosis. After filtering some genes without complete sequence annotation, the binding sites of miR-2008 and genes potentially related to phagocytosis were predicted using the miRanda program. It was found that these four genes (cell division cycle 42, beclin-1, interleukin-1 receptor-associated kinase, and AjElmo1) have potential binding sites for miR-2008 (**Table S2**). We then detected the results of the abundance of the four genes post miR-2008 mimics treatment *in vivo*, and found that the mRNA abundance levels of AjElmo1 and cell division cycle 42 were significantly affected by the miR-2008 mimics, but the other two candidate genes did not change (**Fig. S3**). Further luciferase analysis revealed that miR-2008 mimics could inhibit the luciferase activity of AjElmo1-wt luciferase plasmid, but not the cell division cycle 42-wt luciferase plasmid (**Fig. 5b, Fig. 5c**). Moreover, we detected that AjElmo1 and miR-2008 can be bound by the MS2-RIP assay (**Fig. 5d**). Furthermore, we confirmed that miR-2008 mediates *V. splendidus*-induced coelomocytes phagocytosis through AjElmo1 (**Fig. 5g**). Based on the results of the flow cytometry analysis, we further demonstrated that AjElmo1 positively regulated the coelomocytes phagocytosis in response to *V. splendidus* challenge (**Fig. 6c**). Given that circ432 could interact with miR-2008 and miR-2008 target AjElmo1 and modulate its expression, we thus determined whether circ432 could phagocytosis of coelomocytes by regulating the miR-2008-AjElmo1 axis. The results showed that silencing of circ432 could significantly decrease the mRNA and protein levels of AjElmo1 (**Fig. 7a**). Moreover, the abundance of AjElmo1 in circ432 silenced coelomocytes can be partially restored by adding miR-2008 inhibitors (**Fig. 7b**). More importantly, we detected that the phagocytic ratio in sea

cucumber coelomocytes treated with si-circ432 + miR-2008I + *V. splendidus* was significantly higher than that in its control, sea cucumber coelomocytes treated with si-NC + NCI + *V. splendidus* (**Fig. 7c**). Taken together, we demonstrated that circ432 enhances the coelomocyte phagocytosis via regulating the miR-2008-Elmo1 axis in *V. splendidus*-challenged *Apostichopus japonicus* (**Fig. 8**).

Comment 2: I was surprised that the authors only conducted some simple double luciferase reporter experiments to prove their conclusions. I think that the conclusions reached by only a little single experiment are completely unreliable.

Response: Thanks for the reviewer's professional and valuable comments. Based on your kind and professional suggestions, we supplemented some necessary experiments to draw our conclusions at the RNA level, cell level or protein level. For proving circ432 functions as a miR-2008 sponge, we first confirmed that circ432 could be enriched by the Ago2-RIP assay (**Fig. 3b, newly supplemented results**). Subsequently, we not only proved their binding through the double luciferase report assay with different bind sites or different miR-2008 concentrations (**Fig. 3c, newly supplemented results**), but also displayed that circ432 and miR-2008 are co-located in the cytoplasm through the RNA FISH assay (**Fig. 3d**). For verification AjElmo1 interactions with miR-2008, the results of the double luciferase report assay showed that miR-2008 could bind with AjElmo1 in a dose-dependent manner (**Fig. 5c, newly supplemented results**). Moreover, MS2-RIP assays were also performed to test the direct interaction between miR-2008 and AjElmo1. The results revealed that miR-2008 could be effectively enriched post transfection of pcGFP-MS2-AjElmo1 plasmids in Hela cells, while this result can't be detected in cells transfected with pcGFP-MS2-AjElmo1-mut plasmids (**Fig. 5d, newly supplemented results**). Furthermore, we also detected that transfection of miR-2008 mimics in coelomocytes could inhibit the mRNA and protein levels of AjElmo1 (**Fig. 5e**), while transfection of a miR-2008 inhibitor can significantly improve the mRNA and protein levels of AjElmo1 (**Fig. 5f**). To further prove the regulatory relationship among the circ432-miR-2008-AjElmo1 axis, we detected that silencing of circ432 could

significantly decrease the mRNA and protein levels of AjElmo1, but did not affect the expression level of miR-2008 (**Fig. 7a**). Additionally, we detected that the mRNA and protein levels of AjElmo1 in circ432-silenced coelomocytes can be partially restored by adding miR-2008 inhibitors (**Fig. 7b**). Thus, we confirmed our results can support our conclusions that circ432 enhances the coelomocyte phagocytosis via regulating the miR-2008-Elmo1 axis in *V. splendidus*-challenged *Apostichopus japonicus*.

Comment 3: I did not see any *V. splendidus* infection experiments in this manuscript, but the author repeatedly emphasized in the abstract that their research was carried out under the condition of *V. splendidus* infection, which is very contradictory.

Response: We appreciate the reviewer's comments and are really sorry for our unclear description. Most of our experiments were carried out under the condition of 10^7 CFU/mL *V. splendidus* infection. We detected that circ432 and AjElmo1 were significantly induced in coelomocytes with a 7.2-fold and 6.91-fold post *V. splendidus* challenge (**Fig. 2b, Fig. 6a**). In addition, we used flow cytometry to detect the changes in phagocytosis levels post treated with si-circ432 + *V. splendidus* (**Fig. 2d**), miR-2008 mimics + *V. splendidus* (**Fig. 4b**), miR-2008 inhibitor + *V. splendidus* (**Fig. 4c**), si-AjElmo1 + miR-2008 mimics + *V. splendidus* (**Fig. 5g**), si-AjElmo1 + *V. splendidus* (**Fig. 6c**), or si-circ432 + miR-2008 inhibitor + *V. splendidus* (**Fig. 7c**). The results showed that circ432 as an endogenous sponge RNA interacted with miR-2008 and promoted the abundance of AjElmo1, thereby enhancing the phagocytosis of sea cucumber coelomocytes induced by *V. splendidus*. Furthermore, to facilitate readers' reading and understanding, we modified several captions in the manuscript to highlight the *V. splendidus* infection condition (**Line 780, Line 807-808, Line 845-847**).

Comment 4: Most of the pictures in this manuscript are ambiguous.

Response: Thanks for the reviewer's nice reminder. We have modified the size and resolution of all figures according to the requirements of the journal instruction, and provided the original image in our resubmitted manuscript.

Reviewer 2#

In this study, Fu et al. identified circ432, an intergenic-type circRNA from *A. japonicus*, involved in regulating pathogen-induced coelomocyte phagocytosis via sponge miR-2008 and promoted the abundance of AjElmo1 in sea cucumbers. It was demonstrated that miR-2008 can reduce the translation of AjElmo1 and inhibit AjElmo1-mediated phagocytosis of coelomocytes, which may help *V. splendidus* to evade the host antimicrobial response. The authors further demonstrated that circ432 as an endogenous sponge RNA interacted with miR-2008, and the depressed coelomocytes' phagocytosis by circ432 silencing was consistent with the decreased the abundance of AjElmo1, and could be recovered by miR-2008 inhibitors transfection. This study employed multiple techniques within a combination of *in vivo* and *in vitro* studies to confirm the conclusion. This is a very interesting work on how the circRNA exerts its immune function in a model species that possesses a unique evolutionary status from invertebrate to vertebrate differentiation. Overall, I find that this manuscript is well written and suitable for publication in Communications Biology. However, some critical points need to be addressed to fully support the conclusion.

Response: Thanks for the reviewer's positive comments.

Comment 1: For all the figures, a higher font size and resolution would be desired.

Response: Thanks for the reviewer's nice reminder. We have modified the size and resolution of all figures according to the journal requirements, and also provided the original figures in our resubmitted manuscript.

Comment 2: In Fig. 1A, I don't see whether circ432 is composed of exons or introns and sequence information. The author had better point out the composition of circRNA composed of exons or introns and write the sequence of circ432.

Response: We thank the reviewer for your nice and professional comments and provided the genome structure of circ432 in **Fig. S1**, in which the exon sequence is

shown in red.

Comment 3: In Fig. 1E, the scale should be marked in the micrograph.

Response: We apologize for our negligence and added the scale in the micrograph to **Fig. 1e**.

Comment 4: The author should carefully examine and unify the statistical comparison method and the marking of asterisks used in the whole article. For example, a t-test was used to compare the significant difference between two groups, and the ANOVA was performed among at least three groups or various time points.

Response: Thanks for the reviewer's professional and valuable comments. According to your suggestions, we carefully re-examine and unify the statistical comparison method, and revised the inappropriate making of asterisks used in the article. We marked the "asterisks" by the t-test between two groups and marked the different "letters" by one-way analysis of variance (ANOVA) among at least three groups.

Comment 5: In Fig. 2D, where is the data of the phagocytosis ratio of coelomocytes in the blank group? I did not see the data above the scale on the left?

Response: We apologize for our negligence and add the data of the coelomocyte phagocytosis rate of the blank group in **Fig. 2d**.

Comment 6: In Fig. 5, Fig. 6, and Fig. 7, please show the experimental design briefly and clearly under each band, and it is best to show the band size and the gray value of the target band.

Response: We agree with the reviewer's suggestion and have shown the band size and the gray value of the target band in **Fig. 5e, Fig. 5f, Fig. 6b, Fig. 7a, and Fig. 7b**.

Comment 7: Did you do flow cytometry with cells transfected with circ432 or a construct that has miR-2008 binding sites mutate? Wouldn't this be a more direct test of the effect of circRNA432 on phagocytosis (and potential role in immunity)?

Response: We appreciate the reviewer's professional comments. Actually, we previously tried to transfect with circ432 or a construct that has miR-2008 binding sites mutate into the sea cucumber coelomocytes, unfortunately we failed. In this study, we detected that transfection of circ432 siRNA could significantly decrease the phagocytosis levels of sea cucumbers coelomocyte in response to *V. splendidus* challenge (**Fig. 2d**). Furthermore, the flow cytometry analysis further revealed that co-interference of circ432 and miR-2008 induced higher coelomocytes phagocytosis levels in response to *V. splendidus* challenge, relative to that in sea cucumber coelomocytes treated with si-NC + NCI + *V. splendidus* (**Fig. 7c**). These results also confirmed that circ432 can regulate the coelomocyte phagocytosis induced by the pathogen in *A. japonicus*. Of course, in our future research, we will continue to explore the method of successfully transfecting circRNAs in primary cells. Thanks for your understanding.

Comment 8: siRNA is well known to have off-target effects, while the authors designed two different siRNAs, only one was used for the functional experiments. How the authors confirmed the results could not be due to the off-target effects.

Response: We thank the reviewer for your sincere and valuable comments and add the dose-dependent expression analysis of circ432 after two siRNA transfection in our revised manuscript. The solution to testing whether siRNA has off-target effects is to detect its abundance after interference at the mRNA and protein levels (Wang et al., Pharm Res, 2010, 27(7):1273-1284; Yang et al., Parasitol Res, 2012, 111(3):1251-1257). In this study, two siRNAs of circ432 were designed, and we detected that both circ432 siRNA could significantly decrease their abundance (**Fig. 2c**). Moreover, to better illustrate that siRNAs (si-circ432-1, si-circ432-2) are not off-target, we first detected the abundance of circ432 host gene post si-circ432-1 and si-circ432-2 treatment. The results showed that the abundance level of the host gene of circ432 was not changed (**Fig. 2c**). Moreover, we further examined the effects of the abundance of circ432 post treated with different concentrations of si-circ432-1 and si-circ432-2. The results showed that the expression abundance of circ432 displayed a

trend of decreasing linearly with the increase of siRNA concentration. Furthermore, si-circ432-1 could induce higher inhibition efficiency than si-circ432-2 (**Fig S2**). We thus chose si-circ432-1 for follow-up studies (**Line 169-178**).

Comment 9: In the section of “Materials and methods”, please supplement the description of “western blotting” in the article. Besides, please provide the detail information of all the antibody used in this study.

Response: We apologize for our negligence and we added the description of “**Western blotting**” in the section of “Materials and Methods”, and provided details of all antibodies used in this section. (**Line 520-534**)

Comment 10: In the section of “Prediction of the complementarity among circ432, miR-2008 and AjElmo1” (line 188), did the transcriptome data of circRNAs, miRNAs, and mRNAs in this article uploaded? Please show the accessible ID of the dataset.

Response: Thanks for the reviewer’s sincere comments. The transcriptome data of circRNAs (BioProject ID: PRJNA512578) and mRNA (BioProject ID: SRA080354) has been uploaded and the accessible ID of the dataset was supplemented in the revised manuscript (**Line 453-456**). The necessary transcriptome data information of miRNAs could be founded in the articles (Li et al., 2012, Fish Shellfish Immuno. 33, 436-441).

Comment 11: The database used in the article must be uploaded and accessible. The access number of circ432 and AjElmo1 sequence should be uploaded.

Response: We appreciate the reviewer’s comments. The access number of the circ432 sequence is *Apostichopus* OP242379 (**Line 137**), and the access number of the AjElmo1 sequence is *Apostichopus* OP218937 (**Line 279**).

Reviewer 3#

This study aimed at unravelling the phagocytic molecular mechanism regulated by AjElmo1 under circ432 control as a ceRNA for miR-2008 in sea cucumber

coelomocytes. The authors based this study on their own preliminary data and elsewhere. The research is innovative and the data are original. The study follow a logical workflow and uses several complementary techniques including qPCR, sequencing, flow cytometry, dual luciferase reporter, bioinformatic and protein evaluation.

Response: Thanks for the reviewer's positive comments.

Comment 1: The authors have chosen to use Sybr Green for RT-qPCR. However, there is no indication in the manuscript about validation of the methodology according to the MIQE guidelines (Bustin et al., Clin Chem. 2009 Apr;55(4):611-22. doi: 10.1373/clinchem.2008.112797). For instance, there is no controls to demonstrate no genomic amplification or no specific amplification by the technique. In addition, there is no mention of the validation of housekeeping genes used to normalised the relative quantification.

Response: We appreciate the reviewer's sincere and professional comments. We have supplemented the instructions for qPCR validation according to the MIQE guidelines, which including sample collection, nucleic acid quality control, reverse transcription, validation of housekeeping genes and internal reference gene for relative quantitative standardization, and data analysis methods (**Supplemental Information: MIQE support information and supplementary methods and tables**).

Comment 2: Coelomocytes have the capacity to aggregate. This could be a challenge for accurately count the number of cells between treatment and also normalize the number of cells for RNA, protein, and cell count. How did the author overcome this challenge?

Response: Thanks for the reviewer's sincere and professional comments. As we described in the section on "Primary coelomocytes culture and LPS exposure", we added the precooled anticoagulant (0.02 M EGTA, 0.48 M NaCl, 0.019 M KCl, 0.068 M Tris-HCl, pH 7.6) immediately when collecting the coelomic fluid of sea cucumber (**Line 418-421**). This method has been reported in many previous studies

(Xing et al, *Invertebr Biol*, 1998, 117, 13-22; Cui et al. *Molecular Immunology*, 2018, 479-487; Zhang et al. *Fish Shellfish Immun*, 2014, 383-388). More importantly, as shown in the figure below, we previously observed that the coelomocytes were dispersed when anticoagulants were added, while the coelomocytes without anticoagulants were agglutinated. Based on this, we believe that the number of cells was accurately counted, and thus also normalize the number of cells for RNA, protein, and cell count. Thanks again for the reviewer's careful and professional comments.

Figure legend: Sea cucumber coelomocytes were observed with anticoagulant or not under a microscope (20 ×). Scale bar: 10 µm.

We have done our best to improve our manuscript and made changes wherever necessary. These changes do not influence the content and framework of our paper. We thank the Reviewers for their time and hope that the revised manuscript can meet with your approval.

Looking forward to your favorable decision.

Best wishes,

Sincerely yours,

Ming Guo, Chenghua Li

818 Fenghua Road,

Ningbo University,

Ningbo, Zhejiang Province 315211, P. R. China

Email: guoming@nbu.edu.cn; lichenghua@nbu.edu.cn

Reviewers' comments:

Reviewer #1 (Remarks to the Author):

I don't think the author has solved my concerns about this manuscript. The author hasn't made great changes or added some key experiments. In the revised version, I found that the author only added an unimportant Luc experiment and FISH experiment, which is amazing to me. The author did not rearrange the experimental logic and experimental design of the whole manuscript. In addition, I was very surprised about *V. S* appearing in figure and manuscript. I want to know what this is. If *V. S* represents *V. splendidus*, I can only think that there is a lack of basic standardization of this scientific research. Even the basic standard writing format of species name has problems, and there is a lack of basic respect for scientific research.

Reviewer #2 (Remarks to the Author):

The revised manuscript has addressed all my concerns; it is now acceptable and qualified for publication.

Reviewer #3 (Remarks to the Author):

I would like to acknowledge the authors adding their qPCR validation steps as a supplementary data. However I am still not convinced that the housekeeping genes have been adequately validated. It is well established that multiple housekeeping has to be validated and the less variable expressed genes should be selected for accurate gene expression relative quantification. Please review this paper Vandesompele, J., De Preter, K., Pattyn, F., Poppe, B., Van Roy, N., De Paepe, A., and Speleman, F. (2002). Accurate normalization of real-time quantitative RT-PCR data by geometric averaging of multiple internal control genes. *Genome Biol* 3.

Dear Editor and Reviewers,

Thank you very much for your letter and the comments about our paper “circRNA432 enhances the coelomocyte phagocytosis via regulating the miR-2008-Elmo1 axis in *Vibrio splendidus*-challenged *Apostichopus japonicus* (COMMSBIO-22-2620A), which helped improve our paper’s quality. We have considered the comments and corrected them carefully. We also proofread our manuscript very closely for mistakes and grammatical errors. Here we submit a revised manuscript with color-coded as well as detailed responses. We appreciate your warm work earnestly, and hope that the revised manuscript will meet with approval. The following is our response to each comment point by point.

Reviewer #1:

I don't think the author has solved my concerns about this manuscript. The author hasn't made great changes or added some key experiments. In the revised version, I found that the author only added an unimportant Luc experiment and FISH experiment, which is amazing to me.

Response: Thanks for the reviewer’s comments. In this study, we firstly detected whether circ432 is an exact circRNA by the Sanger sequencing, RNase R treatment, and divergent primers (**Fig.1a, Fig. 1b, Fig. 1c**), and further confirmed that it was significantly induced in coelomocytes post *V. splendidus* challenge and LPS stimulation (**Fig. 2b**), suggesting that circ432 played a vital role in mediating pathogen invasion. Next, we indeed demonstrated that circ432 was a positive regulator involved in *V. splendidus*-induced sea cucumber coelomocyte phagocytosis (**Fig. 2d**). For proving circ432 functions as a miR-2008 sponge, we first detected that circ432 could be enriched by the Ago2-RIP assay (**Fig. 3b**). Subsequently, we not only proved their binding through the double luciferase report assay with different bind sites or different miR-2008 concentrations (**Fig. 3c**), but also displayed that circ432 and miR-2008 were co-located in the cytoplasm through the RNA FISH assay (**Fig. 3d**). Moreover, we

demonstrated that miR-2008M and miR-2008I significantly decreased and increased the phagocytosis ratio of coelomocytes induced by *V. splendidus*, respectively (**Fig. 4d**). For verification AjElmo1 interactions with miR-2008, the results of the double luciferase report assay showed that miR-2008 could bind with AjElmo1 in a dose-dependent manner (**Fig. 5c**). Moreover, MS2-RIP assays were also performed to test the direct interaction between miR-2008 and AjElmo1. The results revealed that miR-2008 could be effectively enriched post transfection of pcGFP-MS2-AjElmo1 plasmids in HeLa cells, while this result could't be detected in cells transfected with pcGFP-MS2-AjElmo1-mut plasmids (**Fig. 5d**). Furthermore, we also detected that transfection of miR-2008 mimics in coelomocytes could inhibit the mRNA and protein levels of AjElmo1 (**Fig. 5e**), while transfection of a miR-2008 inhibitor could significantly improve the mRNA and protein levels of AjElmo1 (**Fig. 5f**). Of course, AjElmo1 was also proved to regulate *V. splendidus*-induced coelomocytes phagocytosis (**Fig. 6**). To further prove the regulatory relationship among the circ432-miR-2008-AjElmo1 axis, we detected that silencing of circ432 could significantly decrease the mRNA and protein levels of AjElmo1, but did not affect the expression level of miR-2008 (**Fig. 7a**). Additionally, we detected that the mRNA and protein levels of AjElmo1 in circ432-silenced coelomocytes can be partially restored by adding miR-2008 inhibitors (**Fig. 7b**). Thus, we confirmed our results could support our conclusions that circ432 enhances the coelomocyte phagocytosis via regulating the miR-2008-Elmo1 axis in *V. splendidus*-challenged *Apostichopus japonicus*.

Comment 1: The author did not rearrange the experimental logic and experimental design of the whole manuscript.

Response: We appreciate the reviewer's comments. As our first round response, we first described why we selected circ432 for this study (**Fig. 1**), characterized of circ432 in sea cucumber coelomocytes (**Fig. 1**) and confirmed the immune function (coelomocyte phagocytosis) it regulated under *V. splendidus* challenge (**Fig. 2**); Since

circRNAs were predominantly located in the cytoplasm and usually function as miRNA sponges, we thus predicted whether circ432 could be bound with the 7 miRNAs that related to SUS-diseased sea cucumbers in our previous studies and selected circ432 and miR-2008 for follow-up research due to their highest possibility, and further demonstrated that circ432 functioned as a miRNA sponge of miR-2008 (**Fig. 3**); Therefore, we then determined whether miR-2008 mediated *V. splendidus*-induced coelomocytes phagocytosis (**Fig. 4**). Next, we screened the potential target genes of miR-2008, confirmed their binding between miR-2008 and AjElmo1, and demonstrated that miR-2008 targeted AjElmo1 to inhibit coelomocytes phagocytosis in sea cucumber (**Fig. 5**). Moreover, we detected that AjElmo1 positively regulated the phagocytosis of *V. splendidus* by sea cucumber coelomocytes (**Fig. 6**). Given that circ432 could interact with miR-2008 and miR-2008 target AjElmo1 and modulate its expression, we further proved the regulatory relationship among the circ432-miR-2008-AjElmo1 axis (**Fig. 7**). Thus, we believed that the experimental logic and experimental design of the whole manuscript was reasonable and convincing. Moreover, the experimental design and experimental logic of this manuscript were similar to others circRNA-miRNA-mRNA researches (Wang et al., 2021, Mol Cancer. 20, 1-20; Zhang et al., 2019, Mol Cancer. 18, 18-20; He et al., 2017, J Exp Clin Canc Res. 36, 145; Bai et al., 2018, J Exp Clin Canc Res. 37, 172). Additionally, we hope that the reviewer can put forward specific comments for revision, we would be happy to revise this manuscript according to your suggestions.

Comment 2: In addition, I was very surprised about V. S appearing in figure and manuscript. I want to know what this is. If V. S represents *V. splendidus*, I can only think that there is a lack of basic standardization of this scientific research. Even the basic standard writing format of species name has problems, and there is a lack of basic respect for scientific research.

Response: We appreciate the reviewer's scientific reminder. We are sorry for the

unclear definition of *V. s* represents *V. splendidus*. We have modified all the *V. s* in the whole revised manuscript to *V. splendidus*, including all the figures, and this really help us.

Reviewer #2:

The revised manuscript has addressed all my concerns; it is now acceptable and qualified for publication.

Response: Thanks very much for the reviewer's positive comments.

Reviewer #3:

I would like to acknowledge the authors adding their qPCR validation steps as a supplementary data. However I am still not convinced that the housekeeping genes have been adequately validated. It is well established that multiple housekeeping has to be validated and the less variable expressed genes should be selected for accurate gene expression relative quantification. Please review this paper Vandesompele, J., De Preter, K., Pattyn, F., Poppe, B., Van Roy, N., De Paepe, A., and Speleman, F. (2002). Accurate normalization of real-time quantitative RT-PCR data by geometric averaging of multiple internal control genes. *Genome Biol* 3.

Response: Thanks very much for the reviewer's professional and constructive comments. According to your valuable suggestions, we validated the expression stability of various housekeeping genes in sea cucumber under different experimental conditions and in different tissues as follows. We first selected ten candidate reference genes based on the frequently used housekeeping genes and genes that showed stable expression in our acquired RNA-seq transcriptome data (Zhang et al., 2014, *Fish Shellfish Immunol.* 38, 383–388), including β -actin, RPS19, RPS18, GAPD, NADH, TUBA, TUBB, NDUFA13, RPL18A, and EF1 α . Upon successfully validating the qRT-PCR specificity and efficiency of these ten primers (**Supplemental Information 1, Fig. 1**), we measured their expression stability using four algorithms (geNorm (Hellemans et al., 2007, *Genome Biol* 8:R19.), NormFinder (Andersen et al., 2004,

Cancer Res 64:5245–5250.), BestKeeper (Pfaffl et al., 2004, Biotechnol Lett 26:509–515.), and RefFinder (<https://blooge.cn/RefFinder/?type=reference>) in *V. splendidus*-challenged *A. japonicus* coelomocytes at five different time points (0, 12, 24, 48 and 72 h) and five healthy *A. japonicus* tissues, including coelomocytes, intestines, muscles, tentacles, and respiratory trees with three biological replicates. According to geNorm algorithm analysis, stability values (M) below 0.5 were considered stable. For *V. splendidus*-challenged sea cucumber coelomocytes, geNorm determined that β -actin and EF1 α were the most stable reference genes, followed by NADH, NDUFA13, TUBB, RPS9, RPS18, and RPL18A (**Supplemental Information 1, Fig. 3a**). In healthy sea cucumber tissues, geNorm determined that β -actin and NDUFA13 were the only two stable reference genes with M below 0.5 (**Supplemental Information 1, Fig. 3b**). According to NormFinder algorithm analysis, the housekeeping gene with the lowest M was the most stable gene. NormFinder identified that β -actin was the most stable reference gene in *V. splendidus*-challenged sea cucumber coelomocytes (**Supplemental Information 1, Fig. 4a**), while RPS18 was the most stably expressed in healthy sea cucumber tissues, followed by β -actin (**Supplemental Information 1, Fig. 4b**). According to BestKeeper algorithm analysis, the expression of β -actin was the most stable reference genes under *V. splendidus* infection, while RPS18 was the most stable reference genes in healthy sea cucumber tissues (**Supplemental Information 1, Table 2**). According to the integration of the three algorithm results by **RefFinder**, β -actin, EF1 α , NDUFA13, and TUBB were the most stable four reference genes in sea cucumber coelomocytes under *V. splendidus* infection (**Supplemental Information 1, Fig. 5a**), while RPS18, β -actin, TUBA, and NDUFA13 were the most stable four reference genes in five different tissues of healthy sea cucumbers (**Supplemental Information 1, Fig. 5b**). Collectively, these results revealed that β -actin could be selected as a better-suited housekeeping gene in tissues of healthy sea cucumbers and in sea cucumber coelomocytes after *V. splendidus* challenge, and the qRT-PCR data with β -actin as the housekeeping gene was convincing in this study.

Detailed information on the stability expression validation of the housekeeping genes was shown in the Supplementary materials (**Supplemental Information 1: Selection of reference genes for qRT-PCR analysis of gene expression in *Apostichopus japonicus* during *Vibrio splendidus* infection**).

We have done our best to improve our manuscript and made changes wherever necessary. These changes do not influence the content and framework of our paper. We thank the Reviewers for their time and hope that the revised manuscript can meet with your approval.

Sincerely,

Chenghua Li

818 Fenghua Road,

Ningbo University,

Ningbo, Zhejiang Province 315211, P. R. China

Email: lichenghua@nbu.edu.cn

REVIEWERS' COMMENTS:

Reviewer #3 (Remarks to the Author):

The authors have addressed my concern.

Dear Reviewers,

Thank you very much for your letter and the comments about our paper “circRNA432 enhances the coelomocyte phagocytosis via regulating the miR-2008-ELMO1 axis in *Vibrio splendidus*-challenged *Apostichopus japonicus* (COMMSBIO-22-2620B), which helped improve our paper’s quality. We have considered the comments and corrected them carefully. We also proofread our manuscript very closely for mistakes and grammatical errors. Here we submit a revised manuscript with color-coded as well as detailed responses. We appreciate your warm work earnestly, and hope that the revised manuscript will meet with approval. The following is our response to each comment point by point.

Reviewer #3:

Comment: The authors have addressed my concern.

Response: Thanks for the reviewer’s positive comments.

We have done our best to improve our manuscript and made changes wherever necessary. These changes do not influence the content and framework of our paper. We thank the Reviewers for their time and hope that the revised manuscript can meet with your approval.

Sincerely,

Chenghua Li

818 Fenghua Road,

Ningbo University,

Ningbo, Zhejiang Province 315211, P. R. China

Email: lichenghua@nbu.edu.cn